# HDFlow: Hierarchical Diffusion-Flow Planning for Long-horizon Robotic Assembly

## Abstract

Long-horizon manipulation tasks represent a significant challenge in robotics, demanding both strategic, high-level reasoning and fast, precise, low-level control. While recent advances in generative models have shown promise in generating behavior plans for long-horizon tasks, they often lack a principled framework for hierarchical decomposition and struggle with the computational demands of real-time execution, due to their iterative denoising process. In this work, we introduce **Hierarchical Diffusion-Flow** (`HDFlow`), a novel hierarchical planning framework that optimally leverages the strengths of *diffusion* and *rectified flow* models. `HDFlow` employs a high-level diffusion planner to generate sequences of strategic subgoals in a learned latent space, capitalizing on diffusion's powerful exploratory capabilities. These subgoals then guide a low-level rectified flow planner that generates smooth and dense trajectories, exploiting the speed and efficiency of ordinary differential equation (ODE)-based trajectory generation. This hybrid approach synergistically combines the strengths of both models to overcome the limitations of single-paradigm generative planners, enabling robust and efficient long-horizon planning. We evaluate `HDFlow` on four challenging furniture assembly tasks in both simulation and real-world, where it significantly outperforms state-of-the-art methods. Project website: https://hdflow-page.github.io/

## 1 Introduction

Robotic manipulation for complex, long-horizon tasks such as robotic assembly Kimble et al. (2020); Suárez-Ruiz & Pham (2016); Lee et al. (2021); Heo et al. (2025); Ankile et al. (2024a;b) remains a significant challenge requiring not only understanding multi-stage instructions and spatial relationships but also executing precise, contact-rich motions over extended periods. Traditional planning methods struggle with long-horizon problems because small inaccuracies in state estimation, dynamics prediction, or control execution accumulate over time, compounding into significant deviations that ultimately lead to task failure. This has motivated a shift towards hierarchical planning (Sacerdoti, 1974; Knoblock, 1990; Singh, 1992; Kaelbling & Lozano-Pérez, 2011), which decomposes a complex goal into a sequence of simpler, more manageable subgoals. A powerful approach within this paradigm is to perform planning in the latent space of a learned world model (Ha & Schmidhuber, 2018; Hafner et al., 2019; 2020). By forecasting future states in a compressed representation, world models allow planners to reason efficiently and abstract away from high-dimensional, noisy observations (Hafner et al., 2022; Wang & Ba, 2020). However, standard world models, trained primarily on reconstruction and dynamics prediction, do not guarantee that the learned latent space is semantically structured for planning. The distance between states in this space does not correlate with progress toward a goal, making it difficult for a planner to navigate effectively.

The advent of generative models, particularly denoising diffusion models (Sohl-Dickstein et al., 2015; Ho et al., 2020; Song et al., 2021), have achieved strong results across various domains (Nichol et al., 2022; Luo & Hu, 2021; Li et al., 2022; Gupta et al., 2024; Avdeyev et al., 2023). Building upon these successes, they have recently revolutionized planning in robotics (Janner et al., 2022; Ajay et al., 2023; Lu et al., 2025b). By treating planning as a conditional generation problem, these models can produce diverse and high-quality trajectories. However, their iterative denoising process is computationally intensive (Dong et al., 2024), making them ill-suited for the fast, low-level control required for real-time robotic interaction. Applying diffusion models naively at all levels of a hierarchy (Chen et al., 2024; Li et al., 2023; Hao et al., 2025) inherits this critical drawback,

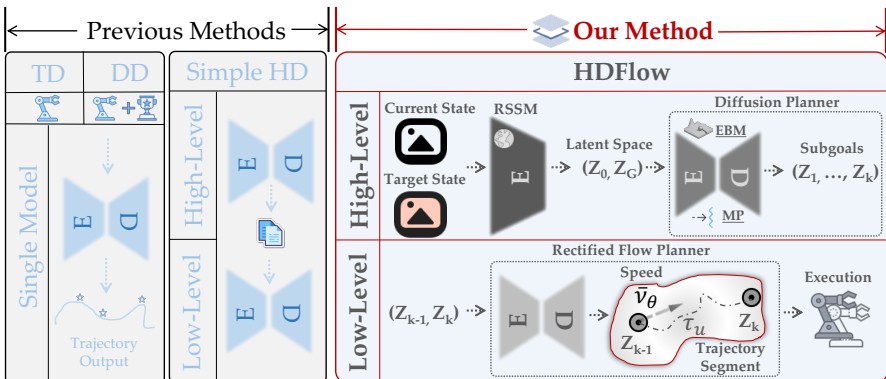

Figure 1: Overview of **HDFlow**: a hybrid hierarchical planning framework for long-horizon robotic assembly. Compared to previous methods such as Traditional Diffuser (TD) and Decision Diffuser (DD) which use a single generative model, and Simple Hierarchical Diffuser (Simple HD) which uses diffusion models at both high and low levels, **HDFlow** employs a high-level diffusion planner for strategic subgoal generation and a low-level rectified flow planner for efficient trajectory synthesis. This hybrid architecture optimally leverages the strengths of both generative paradigms for robust and efficient planning.

creating a bottleneck at the trajectory generation stage. This raises a fundamental question: *Is a single generative modeling paradigm optimal for all levels of a planning hierarchy?*

We empirically show that the answer is no. The requirements for high-level strategic planning are fundamentally different from those of low-level trajectory generation. High-level planning demands exploration and multi-modal diversity to discover viable sequences of subgoals. In contrast, low-level planning demands speed, precision, and deterministic execution to translate a chosen subgoal into a smooth, dense trajectory.

In this paper, we introduce **H**ierarchical **D**iffusion-**F**low (**HDFlow**), a new hierarchical framework for long-horizon manipulation that is built on this core insight. **HDFlow** leverages a hybrid generative architecture that assigns the right tool to the right job. For high-level, strategic planning, we employ a *diffusion model* to generate diverse sequences of subgoals within a learned latent space of a world model. While standard conditional diffusion models can generate plans that are consistent with the goal, they lack an explicit mechanism to assess the quality or long-term viability of those plans. In complex, sparse-reward settings, many plausible-looking sequences of subgoals can lead to irreversible failures. To address this, we introduce an *energy-based model* (EBM) to provide explicit guidance. The EBM is trained to assign low energy to successful strategies and high energy to failing ones, effectively learning a dense reward signal. This energy function then steers the diffusion planner away from potential dead ends and towards high-quality solutions, which is critical for robust, long-horizon performance. Furthermore, to mitigate the failures caused by inaccurate guidance, we enhance the standard EBM guidance with a two-step *manifold-aware* process (Lee & Choi, 2025). For low-level, tactical planning, we introduce a *rectified flow model* to rapidly generate dense latent trajectories to reach each subgoal. The underlying world model itself is trained with a *contrastive objective* to create a semantically structured latent space, which facilitates more effective planning by organizing representations of intermediate states from successful episodes closer to the final goal and pushing them away from failure cases, creating a smoother, more monotonic representation of task progress that is crucial for effective long-horizon planning.

Furthermore, while prior work has been limited to simple tasks with very short-horizons like Maze2D, and AntMaze (Fu et al., 2020), we evaluate our proposed method on four contact-rich assembly tasks from the FurnitureBench (Heo et al., 2025) benchmark in both simulation and real-world settings, including tasks with very long horizons of upto ∼1500 timesteps, involving 11 phases, and assemblies of up to 4 parts to be precisely grasped, oriented, and inserted.

In summary, our contributions are as follows:

- We propose **HDFlow**, a novel, hybrid hierarchical planner that combines a diffusion model for high-level exploration and a rectified flow model for low-level trajectory generation.

- We introduce a two-stage training process featuring a contrastive-trained world model for structured representation learning and a manifold-aware EBM for explicit guidance of the high-level planner, enabling robust planning in sparse-reward environments.

- We demonstrate state-of-the-art performance on four challenging tasks from the FurnitureBench benchmark in both simulation and real-world settings.

## 2 RELATED WORKS

Recent advancements in generative models, particularly denoising diffusion probabilistic models (Sohl-Dickstein et al., 2015; Ho et al., 2020; Song et al., 2021; Karras et al., 2022), have significantly impacted planning in robotics by framing it as a conditional generation problem. Early works such as **Diffuser** (Janner et al., 2022) leverage iterative denoising to generate flexible trajectories conditioned on objectives like rewards or constraints. Building on this, Decision Diffuser (**DD**)(Ajay et al., 2023) further demonstrated that return-conditional diffusion models can outperform traditional offline reinforcement learning by enabling conditioning on various factors like constraints and skills. While powerful, the iterative nature of diffusion models can be computationally intensive (Dong et al., 2024). To tackle long-horizon tasks, hierarchical planning with diffusion models has emerged, as seen in (Li et al., 2023; Chen et al., 2024; Hao et al., 2025). Simple Hierarchical Diffuser (**SHD**)(Chen et al., 2024) employs a high-level diffusion model for sparse subgoal generation and a low-level one for dense trajectory refinement, aiming for improved efficiency and generalization. However, their reliance on diffusion models for both high-level and low-level planning increases the maintenance burden and makes them computationally expensive.

## 3 PRELIMINARIES

**Notation disambiguation.** We use $t$ for environment time indices, $\ell \in \{1, \ldots, L\}$ for diffusion timesteps, and $u \in [0, 1]$ for the continuous flow time in rectified flow. We also denote a clean (noise-free) latent subgoal sequence by $z^{\text{clean}}$ to avoid confusion with the initial environment state $z_0$.

### 3.1 DENOISING DIFFUSION MODELS

Denoising diffusion models (Sohl-Dickstein et al., 2015; Ho et al., 2020) are generative models that learn a data distribution $p(\mathbf{x})$ by reversing a fixed forward noising process.

**Forward Process.** The forward process gradually adds Gaussian noise to a data sample $\mathbf{x}_0$ over $L$ discrete timesteps, according to a variance schedule $\beta_\ell$:

$$q(\mathbf{x}_\ell|\mathbf{x}_{\ell-1}) = \mathcal{N}(\mathbf{x}_\ell; \sqrt{1-\beta_\ell}\mathbf{x}_{\ell-1}, \beta_\ell\mathbf{I}) \tag{1}$$

A key property is that we can sample $\mathbf{x}_\ell$ at any timestep $\ell$ in closed form: $q(\mathbf{x}_\ell|\mathbf{x}_0) = \mathcal{N}(\mathbf{x}_\ell; \sqrt{\bar{\alpha}_\ell}\mathbf{x}_0, (1-\bar{\alpha}_\ell)\mathbf{I})$, where $\alpha_\ell = 1 - \beta_\ell$ and $\bar{\alpha}_\ell = \prod_{i=1}^{\ell} \alpha_i$. As $\ell \to L$, $\mathbf{x}_L$ approaches an isotropic Gaussian distribution $\mathcal{N}(0, \mathbf{I})$.

**Reverse Process.** The reverse process is a learned generative model that starts from noise $\mathbf{x}_L \sim \mathcal{N}(0, \mathbf{I})$ and iteratively denoises it to produce a sample. This process is modeled by a neural network $\epsilon_\theta(\mathbf{x}_\ell, \ell)$ trained to predict the noise $\epsilon$ that was added to the original sample $\mathbf{x}_0$ to produce $\mathbf{x}_\ell$. The training objective is to minimize the mean squared error between the true and predicted noise:

$$\mathcal{L}_{\text{DDPM}} = \mathbb{E}_{\ell, \mathbf{x}_0, \epsilon} \left[ \left\| \epsilon - \epsilon_\theta(\sqrt{\bar{\alpha}_\ell}\mathbf{x}_0 + \sqrt{1-\bar{\alpha}_\ell}\epsilon, \ell) \right\|^2 \right] \tag{2}$$

For conditional generation (e.g., on a context $c$), classifier-free guidance (CFG) (Ho & Salimans, 2022) is commonly used. The model is trained on both conditional and unconditional inputs, and the noise prediction during inference is modified to steer generation towards the context:

$$\hat{\epsilon}_\theta(\mathbf{x}_\ell, \ell, c) = \epsilon_\theta(\mathbf{x}_\ell, \ell, \emptyset) + w \cdot (\epsilon_\theta(\mathbf{x}_\ell, \ell, c) - \epsilon_\theta(\mathbf{x}_\ell, \ell, \emptyset)) \tag{3}$$

where $w$ is the guidance scale and $\emptyset$ denotes the unconditional case. While powerful, the iterative sampling process can be computationally intensive.

## 3.2 RECTIFIED FLOW

Rectified Flow (Lipman et al., 2022; Albergo & Vanden-Eijnden, 2023; Liu et al., 2023) is a generative modeling approach based on ordinary differential equations (ODEs) that offers a more efficient alternative to diffusion models. It learns a deterministic mapping from a simple prior distribution to a data distribution.

Let $p_0$ be a prior distribution (e.g., $\mathcal{N}(0, \mathbf{I})$) and $p_1$ be the data distribution. Rectified Flow constructs straight-line paths between pairs of samples $(\mathbf{x}_0, \mathbf{x}_1)$ drawn from these distributions. The path is defined by the linear interpolation $\mathbf{x}_u = (1 - u)\mathbf{x}_0 + u\mathbf{x}_1$ for $u \in [0, 1]$. The corresponding velocity vector field is simply $\mathbf{x}_1 - \mathbf{x}_0$. The model trains a neural network $v_\theta(\mathbf{x}, u)$ to approximate this vector field by minimizing the flow-matching objective:

$$\mathcal{L}_{RF} = \mathbb{E}_{u, \mathbf{x}_0, \mathbf{x}_1} \left[ \|v_\theta((1 - u)\mathbf{x}_0 + u\mathbf{x}_1, u) - (\mathbf{x}_1 - \mathbf{x}_0)\|^2 \right] \tag{4}$$

Once trained, generation is performed by starting with a sample from the prior, $\mathbf{x}_0 \sim p_0$, and solving the initial value problem for the learned ODE from $u = 0$ to $u = 1$:

$$\frac{d\mathbf{x}_u}{du} = v_\theta(\mathbf{x}_u, u) \tag{5}$$

This is done using a numerical ODE solver. Because it follows a deterministic, straight-line path, Rectified Flow can often generate high-quality samples in significantly fewer function evaluations than required by iterative diffusion models, making it ideal for applications requiring fast synthesis, such as real-time trajectory generation.

## 4 METHOD

In this section, we introduce **HDFlow**, a hierarchical planning framework that tackles long-horizon manipulation tasks using a two-stage training process. First, we train a world model with a contrastive objective to learn a semantically structured latent space that embeds a notion of progress. Second, we train a hierarchical planner on top of the frozen, pre-computed latent representations from this world model. This planner consists of a high-level diffusion model for strategic subgoal generation and a low-level rectified flow model for efficient trajectory synthesis.

### 4.1 STAGE 1: WORLD MODEL LEARNING

Our framework operates within the latent space of a world model, which is trained to model the environment's dynamics from multi-modal, high-dimensional observations (denoted as $o$). We adopt a Recurrent State Space Model (RSSM) architecture (Hafner et al., 2023) with an encoder that leverages a pretrained DINOv2 model (Oquab et al., 2024). The RSSM learns to encode observations into a latent space $\mathcal{Z}$, and is trained to accurately reconstruct observations and predict future states. For a detailed explanation of the architecture and training objective, please see Appendix B.

$$\mathcal{L}_{\text{WM}} = \mathbb{E}_{q_\phi(z_{1:T}|o_{1:T})} \left[ \sum_{t=1}^{T} (\log p_\phi(\hat{o}_t|z_t, h_t) - D_{KL}[q_\phi(z_t|h_t, e_t)||p_\phi(\hat{z}_t|h_t)]) \right] \tag{6}$$

The first term, $\log p_\phi(\hat{o}_t|z_t, h_t)$, is the *observation reconstruction loss*, which ensures that the world model can accurately reconstruct the observed input $o_t$ from its latent representation $z_t$ and recurrent hidden state $h_t$. The second term, $D_{KL}[q_\phi(z_t|h_t, e_t)||p_\phi(\hat{z}_t|h_t)]$, is a *KL divergence regularization term*. It minimizes the divergence between the posterior latent distribution $q_\phi(z_t|h_t, e_t)$ (inferred from the actual observation embedding $e_t$) and the prior latent distribution $p_\phi(\hat{z}_t|h_t)$ (predicted solely from the recurrent hidden state $h_t$). This term acts as a bottleneck, forcing the model to learn compressed, predictive latent states.

While the standard world model objective, $\mathcal{L}_{\text{WM}}$, encourages predictable dynamics, it does not explicitly structure the latent space to reflect a notion of progress towards a goal, making long-horizon planning challenging. To address this, we introduce a contrastive learning objective designed to provide a dense learning signal and organize the latent space for more effective downstream planning. The goal is to pull representations of intermediate latent states from successful trajectories closer to their final goal representation, while pushing them away from intermediate latent states from failed trajectories.

Figure 2: **HDFlow pipeline.** The framework consists of two main stages: **World Model Learning** (left), where observations are encoded into a structured latent space, and **Hierarchical Planner Training** (right). The latter involves a *High-Level* diffusion planner generating sparse strategic subgoals $(z_1, \ldots, z_K)$ with EBM guidance, and a *Low-Level* rectified flow planner synthesizing dense trajectories $\tau = [\tau_1, \ldots, \tau_H]$ between subgoals using an ODE solver.

Let $f(\cdot)$ be a projection head that maps latent states $z$ to a new embedding space. For a given intermediate latent state $z_k$ from a successful trajectory with final goal $z_G$, we form a positive pair $(f(z_k), f(z_G))$. Negative pairs are formed with intermediate latent states from failed trajectories. The contrastive loss is then given by the modified InfoNCE objective (Oord et al., 2018) as suggested by (Schroff et al., 2015; Sohn, 2016):

$$\mathcal{L}_{\text{contrastive}} = -\mathbb{E}\left[\log \frac{\exp(\text{sim}(f(z_k), f(z_G))/\tau)}{\sum_{j=1}^{N} \exp(\text{sim}(f(z_k), f(z_j))/\tau)}\right] \tag{7}$$

where $\text{sim}(\cdot, \cdot)$ is the cosine similarity and $\tau$ is a temperature hyperparameter (Wang & Liu, 2021).

To further encourage the latent space to be informative for control, we also include an inverse dynamics model (Agrawal et al., 2016; Pathak et al., 2018). This model, $a_t \sim p_\phi(a_t | z_t, z_{t+1})$, is trained to predict the action that was taken to transition between two consecutive latent states. It is trained with a mean squared error loss:

$$\mathcal{L}_{IDM} = \mathbb{E}\left[\|a_t - \text{MLP}(z_t, z_{t+1})\|^2\right] \tag{8}$$

This objective ensures that the latent states encode action-relevant information, which is beneficial for the downstream planner. The full training objective combines the world model loss, the inverse dynamics loss, and the contrastive loss:

$$\mathcal{L}_{\text{WM-total}} = \lambda_{WM}\mathcal{L}_{WM} + \lambda_{IDM}\mathcal{L}_{IDM} + \lambda_{\text{contrastive}}\mathcal{L}_{\text{contrastive}} \tag{9}$$

After this stage, the world model's weights are frozen, and its encoder is used to generate a static dataset of structured latent representations for the next stage.

## 4.2 STAGE 2: HIERARCHICAL PLANNER TRAINING

With the structured latent space from Stage 4.1 fixed, we frame the long-horizon planning problem as a conditional generative modeling task. We decompose the problem by defining a temporal abstraction, a common and effective strategy in hierarchical planning (Li et al., 2023; Chen et al., 2024). For each full-length trajectory in our latent dataset, we define subgoals by subsampling the trajectory at a fixed interval of $H$ timesteps. For a trajectory of length $T$ (environment timesteps), this yields $K = \lfloor T/H \rfloor$ subgoals, which can vary across trajectories. This process creates two distinct datasets for our planners:

- For the **high-level planner**, we create sparse sequences of $K$ latent subgoals, $z = (z_1, \ldots, z_K)$, where each subgoal $z_k$ corresponds to the latent state at timestep $k \cdot H$.
- For the **low-level planner**, we create a dataset of dense, fixed-length trajectory segments. Each segment $\tau_k$ contains the $H$ latent states and actions between consecutive subgoals $z_{k-1}$ and $z_k$.

This decomposition allows us to train a high-level planner $\pi_{HL}$ to generate a sequence of latent subgoals, and a low-level planner $\pi_{LL}$ to generate the dense trajectory segment to reach each subgoal.

### 4.2.1 HIGH-LEVEL PLANNER: MANIFOLD-AWARE EBM-GUIDED DIFFUSION

The high-level planner is a conditional diffusion model, trained to generate a sequence of $K$ latent subgoals, $z = (z_1, \ldots, z_K)$, conditioned on the context $c = (z_0, z_G)$, where $z_0$ is the current latent state and $z_G$ is the goal latent state. It is trained by minimizing the standard noise prediction error from Eq. 2:

$$\mathcal{L}_{HL} = \mathbb{E}_{\ell, z^{\text{clean}}, \epsilon} \left[ \left\| \epsilon - \epsilon_\theta \left( \sqrt{\bar{\alpha}_\ell} \, z^{\text{clean}} + \sqrt{1 - \bar{\alpha}_\ell} \, \epsilon, \ell, c \right) \right\|^2 \right] \tag{10}$$

While this provides a strong prior for generating plausible plans, it does not guarantee that all generated plans will be successful, especially in long-horizon scenarios where small errors can compound. To address this and actively steer the planner towards high-quality solutions, we introduce an Energy-Based Model (EBM) for explicit guidance at inference time. The EBM, $E_\phi(z|z_0, z_G)$, is a separate network trained to predict a low energy for high-quality latent subgoal sequences and a high energy for poor ones. It is trained with a contrastive loss that pushes down the energy of plans from successful trajectories ($z_{\text{pos}}$) and pushes up the energy of plans from failed trajectories ($z_{\text{neg}}$):

$$\mathcal{L}_{EBM} = \log(1 + \exp(E_\phi(z_{\text{pos}}) - E_\phi(z_{\text{neg}}))) \tag{11}$$

However, in high-dimensional latent spaces, inexact guidance can cause *manifold deviation* (He et al., 2024), a phenomenon where guided samples drift away from the feasible latent subgoal manifold $\mathcal{M}_t$. We formalize this issue through the guidance gap:

**Definition 1** *Let $\nabla_{z_t} E_{true}(z_t|c)$ denote the true optimal energy guidance and $\nabla_{z_t} E_\phi(z_t|c)$ be our learned EBM guidance. The guidance gap at $z_t$ is:*

$$\Delta_{EBM}(z_t) = \| \nabla_{z_t} E_{true}(z_t|c) - \nabla_{z_t} E_\phi(z_t|c) \|_2 \tag{12}$$

**Proposition 4.1** *(Proof in Appendix) The EBM guidance gap $\Delta_{EBM}(z_t)$ has a lower bound scaling as $\frac{c}{\sqrt{1-\bar{\alpha}_t}} \sqrt{d}$ in high-dimensional latent spaces, where $c > 0$ is a constant independent of dimensionality $d$.*

Definition 1 and Proposition 4.1 show that inaccuracies in energy guidance grow with scenarios involving long planning horizons and high-dimensional latent spaces, leading sampled trajectories to deviate away. Check Appendix D.3 for a detailed proof.

**Manifold-Aware Guidance.** High-dimensional latent representations often exhibit intrinsic low-dimensional structure. Under our contrastive training, successful latent subgoal sequences concentrate on a $k$-dimensional submanifold $\mathcal{M}_0 \subset \mathbb{R}^d$ with $k \ll d$. To mitigate this manifold deviation, we enhance the standard EBM guidance with a two-step manifold-aware process as suggested in (Lee & Choi, 2025). Rather than applying guidance directly to the noise prediction, we perform:

**Step 1: Guided Sampling.**

$$z_{\ell-1}^{\text{temp}} \sim \mathcal{N}\left(\mu_\theta(z_\ell) + w_{ebm} \Sigma^\ell g, \Sigma^\ell\right) \tag{13}$$

where $g = \nabla_{z_\ell} E_\phi(z_\ell|c)$ is the EBM guidance and $\mu_\theta, \Sigma^\ell$ are the mean and covariance of the reverse diffusion transition (scheduler-dependent, e.g., DDPM variance-preserving).

**Step 2: Manifold Projection.**

$$z_{\ell-1} = \mathcal{P}_{\mathcal{T}_{z_{\ell-1}} \mathcal{M}_{\ell-1}}\left(z_{\ell-1}^{\text{temp}}\right) \tag{14}$$

where $\mathcal{P}_{\mathcal{T}_{z_{\ell-1}} \mathcal{M}_{\ell-1}}$ projects onto the local tangent space of the latent manifold $\mathcal{M}_{\ell-1}$.

The manifold projection is computed using local low-rank approximation: we first obtain a denoised estimate using Tweedie's formula (Robbins, 1992; Chung et al., 2022; 2023):

$$\hat{z}^{0|\ell-1} = \frac{1}{\sqrt{\bar{\alpha}_{\ell-1}}} \left( z_{\ell-1}^{\text{temp}} - \sqrt{1 - \bar{\alpha}_{\ell-1}} \, \epsilon_\theta(z_{\ell-1}^{\text{temp}}, \ell-1, c) \right) \tag{15}$$

We then retrieve $k$ nearest neighbors from successful latent subgoal sequences using cosine similarity (Feng et al., 2024), forward diffuse them to timestep $\ell - 1$, and perform rank-$r$ PCA to obtain

the projection basis $U \in \mathbb{R}^{d \times r}$. Let $\mu$ be the local mean of these neighbors. The mean-centered projection is

$$\mathcal{P}(\mathbf{z}) = \mu + UU^T(\mathbf{z} - \mu).$$

**Proposition 4.2** *Given a base diffusion planner $\epsilon_\theta$ trained for classifier-free guidance, a learned energy function $E_\phi(z|c)$, and a manifold projection operator $\mathcal{P}_\mathcal{M}$, the manifold-aware guided planner, which combines EBM guidance with manifold projection, corresponds to sampling from a posterior distribution $p(z|y = 1, z \in \mathcal{M}, c)$ that maximizes the likelihood of generating a successful and feasible goal-conditioned plan.*

### 4.2.2 Low-Level Planner: Rectified Flow for Trajectory Generation

The low-level planner's role is to generate a dense, short-horizon latent trajectory $\tau_z$ to a given subgoal $z_k$. We can frame this subproblem through the lens of optimal transport, which seeks the most efficient way to transform one probability distribution into another. Here, the task is to find the optimal mapping from the distribution of initial states around $z_{k-1}$ to the distribution of target states around $z_k$. The *cost* of transport is minimized by trajectories that are as straight as possible in the latent space. We use a conditional rectified flow model, $v_\theta(\tau_u, u, c_k)$, for its speed. It is trained to generate a trajectory segment $\tau$ conditioned on the context $c_k = (z_{k-1}, z_k)$ by minimizing the standard flow-matching objective from Eq. 4:

$$\mathcal{L}_{LL} = \mathbb{E}_{u,\tau_0,\tau_1} \left[ \left\| v_\theta\big((1 - u)\tau_0 + u\tau_1, u, c_k\big) - (\tau_1 - \tau_0) \right\|^2 \right] \tag{16}$$

**Construction of training pairs.** For each consecutive latent subgoal pair $(z_{k-1}, z_k)$, we extract the dense $H$-step latent segment from successful demonstrations. We set $\tau_1$ to be the observed segment that ends at $z_k$ and $\tau_0$ to be the segment that starts at $z_{k-1}$ from the same demonstration. We optionally apply small Gaussian perturbations in latent space and mild time-warping for robustness; see Appendix B for details.

### 4.2.3 Planner Training Objective

The components of the hierarchical planner are trained jointly with a composite loss function that combines the objectives for the high-level planner, the low-level planner, the EBM, and manifold consistency:

$$\mathcal{L}_{\text{planner}} = \lambda_{HL}\mathcal{L}_{HL} + \lambda_{LL}\mathcal{L}_{LL} + \lambda_{\text{EBM}}\mathcal{L}_{\text{EBM}} + \lambda_{\text{proj}}\mathcal{L}_{\text{projection}} \tag{17}$$

where the $\lambda$ terms are hyperparameters and $\mathcal{L}_{\text{projection}}$ encourages the generated latent subgoals to remain close to the learned latent manifold:

$$\mathcal{L}_{\text{projection}} = \mathbb{E}_{z \sim \pi_{HL}} \left[ \|z - \mathcal{P}_\mathcal{M}(z)\|^2 \right] \tag{18}$$

### 4.2.4 Inference and Online Deployment

During online deployment in an MPC framework, the high-level planner is invoked iteratively. At each replanning step, it takes the robot's *current latent state* (which updates as the low-level planner executes segments) and the *goal state* as input to generate a new sequence of subgoals for the remaining portion of the task. The low-level planner takes the first subgoal from this sequence, and generates a dense latent trajectory. Then consecutive latent pairs $(z_t, z_{t+1})$ along the generated latent trajectory are mapped to control actions via the inverse dynamics model $p_\phi(a_t \mid z_t, z_{t+1})$ introduced in Stage 4.1. Actions are executed for $H$ steps within an MPC loop before replanning with the updated state.

## 5 Experiments

To evaluate the efficacy of our proposed framework, we design a set of experiments to answer the following key questions: (**1**) Does `HDFlow` achieve state-of-the-art **performance** on complex, long-horizon robotic assembly tasks compared to existing methods? (**2**) How does our **hybrid** architecture compare against non-hybrid hierarchical planner as well as single planner approaches? (**3**) What are the contributions of the **core components** of `HDFlow`? We will conduct ablation studies to analyze the impact of our hierarchical structure and planner choices. (**4**) Does `HDFlow` offer superior **computational efficiency** during inference, a critical factor for real-time robotic control?

| Method | one_leg | | | lamp | | | round_table | | | cabinet | |
|---|---|---|---|---|---|---|---|---|---|---|---|
| | Low | Med | High | Low | Med | High | Low | Med | High | Low | Med |
| **BC** | 0 | 0 | 0 | 0 | 0 | 0 | 0 | 0 | 0 | 0 | 0 |
| **DP** | 51 | 19 | 3 | 18 | 7 | 1 | 6 | 2 | 0 | 4 | 1 |
| **JUICER** | 68 | 22 | 3 | 27 | 12 | 2 | 23 | 8 | 2 | 11 | 5 |
| **Diffuser** | 56 | 22 | 4 | 22 | 9 | 1 | 21 | 7 | 1 | 6 | 2 |
| **DD** | 60 | 22 | 4 | 24 | 11 | 1 | 22 | 8 | 2 | 9 | 3 |
| **HDMI** | 66 | 26 | 11 | 37 | 16 | 11 | 33 | 15 | 9 | 17 | 8 |
| **SHD** | 71 | 31 | 15 | 43 | 22 | 16 | 41 | 21 | 12 | 21 | 11 |
| **Ours** | **92** | **71** | **39** | **68** | **49** | **34** | **61** | **43** | **27** | **55** | **36** |

Table 1: Main results on FurnitureBench tasks in simulation. Success rates (%) are reported for different initial randomization levels (**Low**, **Med**, **High**).

| Method | one_leg | lamp |
|---|---|---|
| **FD** | 60 | 24 |
| **HF** | 63 | 24 |
| **HD** | 71 | 43 |
| **Ours** | **92** | **68** |

Table 2: Ablation study on the choice of generative models for the high-level and low-level planners. Success rates (%) are reported for the **one_leg** and **lamp** tasks under **Low** randomization.

| Method | one_leg | lamp |
|---|---|---|
| **Ours** | **92** | **68** |
| **w/o Manifold Projection** | 84 | 57 |
| **w/o Manifold-aware EBM** | 61 | 33 |
| **w/o Contrastive WM** | 58 | 27 |

Table 3: Ablation study on the core components of **HDFlow**. Success rates (%) are reported for the **one_leg** and **lamp** tasks under **Low** randomization.

## 5.1 SIMULATION EXPERIMENTS

**Tasks and Environment.** We evaluate our method on FurnitureBench (Heo et al., 2025), a challenging benchmark for long-horizon, contact-rich robotic assembly. We chose 4 tasks from the benchmark: **one_leg**, **lamp**, **round_table**, and **cabinet** (see Figure 4 for start and goal positions), with the default initial randomization protocol: **Low**, **Med**, and **High**. We define task success as assembling all the furniture parts in their goal poses. We report the success rate calculated over 100 episodes for each task. Further details about the tasks and dataset collection are provided in Appendix A

**Implementation Details.** We adopt Diffusion Transformer (DiT) (Peebles & Xie, 2023) as the backbone for the high-level diffusion planner and Rectified Flow Transformer (Esser et al., 2024) for the low-level rectified flow planner. Detailed descriptions of our experimental setup, including model architectures, training procedures, and hyperparameter settings, are provided in Appendix B

**Baselines.** We compare **HDFlow** against a comprehensive set of state-of-the-art methods that cover different paradigms for long-horizon manipulation. We provide more details about each baseline in the Appendix C.

- **Imitation Learning (IL) Baselines**: These methods learn directly from successful demonstrations without the use of explicit reward signals.(a) Vanilla behaviour cloning (**BC**) (b) Diffusion Policy (**DP**) (Chi et al., 2023) (c) **JUICER** (Ankile et al., 2024a)

- **Diffusion-based Planners**: These methods use diffusion-based planning methods from a mixed dataset of successful and failure demonstrations. (a) **Diffuser** (Janner et al., 2022): A diffusion probabilistic model that plans by iteratively denoising trajectories. It treats planning as a conditional generation problem and can produce diverse and high-quality trajectories. (b) Decision Diffuser (**DD**) (Ajay et al., 2023) (c) **HDMI** (Li et al., 2023) (d) Simple Hierarchical Diffuser (**SHD**) (Chen et al., 2024)

**Analysis.** The results presented in Table 1 demonstrate **HDFlow**'s superior performance across all four challenging furniture assembly tasks and varying levels of initial randomization. Our hybrid, hierarchical planning framework consistently achieves the highest success rates, significantly outperforming imitation learning, non-hierarchical, and other hierarchical diffusion planners.

Figure 3: (left) Real-world FurnitureBench setup. (right) A successful rollout of **HDFlow** planner on the **one_leg** assembly task initialized with **Med** randomness in both simulation (top) and real-world (bottom).

| Method | SR | Inference Time |
|--------|-----|----------------|
| **FD** | 24 | 197 |
| **HF** | 24 | 53 |
| **HD** | 43 | 142 |
| **Ours** | **68** | **88** |

Table 4: Ablation study on computational efficiency for the **lamp** task (**Low** randomization). This table presents both success rates (SR, %) and average inference time (milliseconds per planning step).

| Method | one_leg Low | one_leg Med | lamp Low | lamp Med | round_table Low | round_table Med |
|--------|------|------|------|------|------|------|
| **BC** | 0/10 | 0/10 | 0/10 | 0/10 | 0/10 | 0/10 |
| **IQL** | 3/10 | 1/10 | 1/10 | 0/10 | 0/10 | 0/10 |
| **Ours** | **8/10** | **6/10** | **5/10** | **4/10** | **4/10** | **3/10** |

Table 5: Main results on FurnitureBench tasks in real-world. Success rates (number of successes out of 10 episodes) are reported for different initial randomization levels (**Low**, **Med**).

## 5.2 ABLATION STUDIES

To systematically analyze the contribution of each component within **HDFlow**, we conduct extensive ablation studies. In this section, we consider **one_leg** and **lamp** tasks under **Low** randomization. For more ablation experiments, please see Appendix E

**Choice of Generative Model.** We compare **HDFlow** against variants that use alternative generative models. Specifically, we consider **FD** (flat diffusion), **HF** (hierarchical flow), and **HD** (hierarchical diffusion). Table 2 shows that our planner significantly outperforms single-paradigm or flat planning approaches.

**Comparison of Wall-clock times.** To assess the computational efficiency of **HDFlow** and its variants, we conducted an ablation study measuring the average inference time per planning step. Table 4 presents these results alongside the success rates for the **lamp** task (**Low** randomization), directly building upon the analysis from Table 2.

**Contribution of Core Components.** We investigate the impact of the contrastive world model, EBM guidance, and manifold projection by progressively removing them from **HDFlow**. Table 3 demonstrates that each novel component measurably contributes to **HDFlow**'s overall success, highlighting the importance of our multi-faceted approach to long-horizon robotic assembly.

## 5.3 REAL-WORLD EXPERIMENTS

We conduct experiments in the real-world using a Franka Research 3 robot arm by setting up the benchmark as shown in Figure 3. We finetune our world model and hierarchical planners using 50 real-world demonstrations for each task, and report success rate on 10 evaluation episodes in Table 5. We compare our method with (a) Vanilla Behavior Cloning (**BC**), and (b) Implicit Q-Learning (**IQL**) (Kostrikov et al., 2022). **HDFlow** significantly outperforms both baselines, achieving high success rates on all tasks and maintaining robust performance even under increased initial randomization, which validates its effectiveness for real-world robotic assembly.

## 6 CONCLUSION

In this work, we introduced **HDFlow**, a novel hierarchical planning framework that effectively addresses long-horizon, contact-rich robotic assembly tasks by synergistically combining the strengths of diffusion and rectified flow models. Our approach leverages a contrastively-trained world model to learn a semantically structured latent space, a manifold-aware EBM-guided high-level diffusion planner for strategic subgoal generation, and a fast low-level rectified flow planner for efficient

trajectory synthesis. We demonstrated state-of-the-art performance on challenging FurnitureBench tasks, showcasing the robustness and efficiency of **HDFlow** in both simulation and real-world settings.

**Limitations.** Despite its significant performance, **HDFlow** has several limitations that suggest avenues for future research. Currently, our framework relies on a dataset of successful and failed demonstrations for training the world model and the EBM. While effective, collecting such data can be resource-intensive. Another area for improvement lies in the computational efficiency of the high-level diffusion planner. Although rectified flow handles low-level trajectory generation efficiently, the iterative nature of diffusion can still pose a bottleneck for tasks that require rapid online replanning.

## REPRODUCIBILITY STATEMENT

We recognize the importance of reproducible research in machine learning and robotics. To ensure the reproducibility of our work, we have made comprehensive efforts throughout the development and evaluation of `HDFlow`. We encourage readers to refer to the following sections for details: For our novel hierarchical planning framework and its components, including the high-level diffusion planner and low-level rectified flow planner, their architectural specifics, training procedures, and hyperparameter settings are thoroughly described in Section 4.2 and Appendix B. The mathematical derivations and proofs for our theoretical propositions, including the EBM guidance gap and manifold-aware guided planning, are provided in Appendix D. Regarding the experimental setup and datasets, a complete description of the FurnitureBench tasks, environment configurations, data collection protocols, and action/observation spaces can be found in Section 4.1 and Appendix A. Upon acceptance of this paper, we will publicly release the source code for `HDFlow` and all experimental setups to facilitate further research and validation.

## LLM USAGE STATEMENT

We utilized large language models (LLMs) to polish the presentation of this paper and to assist with grammar and style corrections. All scientific content, experimental design, and analysis remain the sole work of the authors.

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

# A TASKS AND DATASET

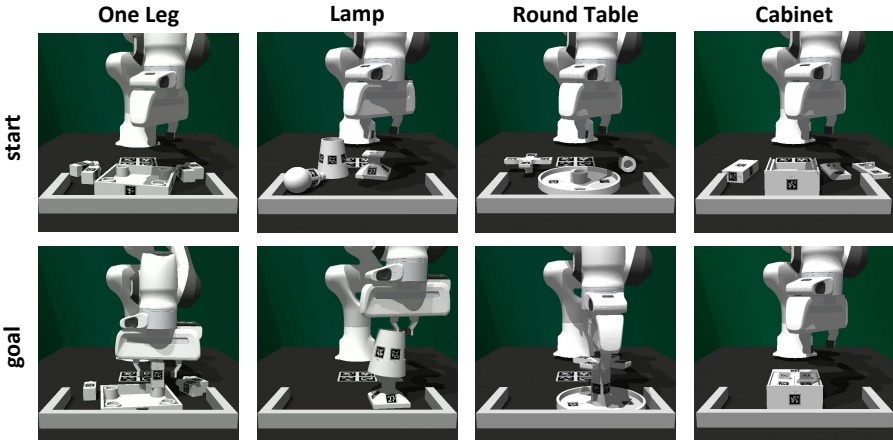

Figure 4: Overview of tasks from FurnitureBench in simulation.

FurnitureBench (Heo et al., 2025) is a novel furniture assembly benchmark for testing complex, long-horizon manipulation tasks. We choose a subset of 4 tasks from the available 9 tasks in the benchmark. The tasks involve assembling various pieces of furniture from individual parts using a simulated Franka Emika Panda robot in a IsaacGym environment. The robot receives multi-modal observations, including front and wrist RGBD images and its proprioceptive state. The goal is to assemble the furniture correctly, which requires a sequence of precise manipulations over a long time horizon as shown in Fig 4. Each task comes with three different levels with respect to the randomness in the initial furniture part configuration, making the manipulation task more challenging.

- **Low**: Furniture parts are randomly offset from their original positions by $[-1.5, 1.5]$ cm in the horizontal plane
- **Med**: Furniture parts are randomly offset from their original positions by $[-5, 5]$ cm and from their original rotations by $[-45°, 45°]$ in the horizontal plane.
- **High**: Furniture parts are randomly initialized on the workspace.

Below we describe each task in detail:

1. **One Leg**
    - **Task**: The task is to assemble one leg of a table. First, it has to stabilize the tabletop in one corner of a U-shaped wall, then it has to grasp, insert and screw the leg in its goal position.
    - **Phases**: 5 phases in total.
    - **Success Metric**: Assemble 2 parts in their goal positions.
    - **Max. Episode Length**: 700
2. **Lamp**
    - **Task**: The task is to assemble one lamp base, bulb, and lamp hood. First, it has to stabilize the lamp base in one corner of a U-shaped wall. Then, it has to grasp, insert and screw the bulb into the lamp base. Finally, it has to grasp, and insert the lamp hood into the bulb.
    - **Phases**: 7 phases in total.
    - **Success Metric**: Assemble 3 parts in their goal positions.
    - **Max. Episode Length**: 1100
3. **Round Table**
    - **Task**: The task is to assemble one round tabletop, leg, and table base. First, it has to stabilize the round tabletop in one corner of a U-shaped wall. Then, it has to grasp,

insert and screw the leg into the tabletop. Finally, it has to grasp, insert and screw the table base into the leg.

- **Phases**: 8 phases in total.
- **Success Metric**: Assemble 3 parts in their goal positions.
- **Max. Episode Length**: 1500

4. **Cabinet**

- **Task**: The task is to assemble one cabinet body, left door, right door, and cabinet top. First, it has to stabilize the cabinet body in one corner of a U-shaped wall, then it has to grasp, insert and slide each door in its goal position. Then, it has to flip the cabinet body along with the assembled doors orthogonally. Finally, it has to grasp, insert and screw the cabinet top.
- **Phases**: 11 phases in total.
- **Success Metric**: Assemble 4 parts in their goal positions.
- **Max. Episode Length**: 1500

**Simulation.** We use a scripted policy provided by the original benchmark to collect 100 successful and 50 failure demonstrations for each task and randomness type. Unfortunately, we were unable to collect any successful trajectories with high randomness initialization for cabinet task, so we only considered low and med initial randmoness.

**Real-world.** Following (Ankile et al., 2024a;b; Ren et al., 2024), we use 3DConnexion Space-Mouse, a 6DoF end-effector control device for teleoperated dataset collection. For each task and randomness type, we collect 50 successful demonstrations and use them to finetune our pretrained world model and hierarchical planners.

**Observation Space.** The observation space for our world model consists of two RGBD images from the front and wrist camera and robot proprioceptive states. The front and wrist camera RGB images are first resized from $1280 \times 720$ to $320 \times 240$, and then center cropped to $224 \times 224$. The robot proprioceptive state consists of current end-effector (EE) state and gripper width. In particular, 3 dimensional EE position, 4 dimensional EE orientation, 3 dimensional linear velocity, 3 dimensional rotational velocity, and 1 dimensional gripper width.

**Action Space.** We use an 8D action space, which consists of 3 dimensional delta EE position, 4 dimensional delta EE orientation (quaternion), and 1 dimensional gripper action. The action space is bounded between $-1$ and $+1$.

**Goal Specification.** We initialize the furniture in its final assembled state and capture the front and wrist RGBD images as well as the robot's proprioceptive state. These are then passed through the pre-trained world model's encoder to produce a fixed latent goal vector $z_G$. This vector is then used as the conditioning for the planner in all subsequent experiments for that task.

## B IMPLEMENTATION DETAILS

### B.1 WORLD MODEL

World models learn a compressed representation of the environment's state and a model of its dynamics within this latent space. We use a Recurrent State-Space Model (RSSM) (Hafner et al., 2019), which has demonstrated strong performance in modeling complex dynamics from high-dimensional observations. We use observations from both front and wrist RGB-D cameras. The RSSM consists of the following components:

- **RGB Encoder:** A visual encoder leveraging a pretrained DINOv2 model (Oquab et al., 2024) for RGB images, mapping them to a lower-dimensional embedding.

- **Depth Encoder:** A separate CNN that encodes depth images into an embedding.

- **State Encoder:** An MLP that encodes the robot's proprioceptive state into an embedding.

- **Combined Observation Embedding:** The embeddings from the RGB, Depth, and State encoders are concatenated to form the high-dimensional observation embedding $e_t = \text{Enc}_\phi(o_t)$.

- **Dynamics Model:** The dynamics are modeled in two parts. A deterministic RNN, $h_{t+1} = f_\phi(h_t, z_t)$, updates its hidden state to summarize the history. This hidden state is then used to predict a prior distribution over the current latent state, $\hat{z}_t \sim p_\phi(\hat{z}_t|h_t)$.

- **Representation Model:** A posterior distribution over the latent state $z_t \sim q_\phi(z_t|h_t, e_t)$ is inferred from the combined observation embedding $e_t$ and the deterministic hidden state $h_t$ of the RNN.

- **Decoder:** A decoder $\hat{o}_t \sim p_\phi(\hat{o}_t|h_t, z_t)$ reconstructs the original observation from the latent state.

The model is trained by maximizing the Evidence Lower Bound (ELBO) on the data log-likelihood, which encourages accurate reconstruction and prediction while regularizing the latent space. The objective function is:

$$\mathcal{L}_{\text{WM}} = \mathbb{E}_{q_\phi(z_{1:T}|o_{1:T})} \left[ \sum_{t=1}^{T} \left( \log p_\phi(\hat{o}_t|z_t, h_t) - D_{KL}[q_\phi(z_t|h_t, e_t)||p_\phi(\hat{z}_t|h_t)] \right) \right] \tag{19}$$

This objective trains the model to form a compressed and predictive latent space $\mathcal{Z}$, where planning can be performed efficiently.

| Parameter | Value |
|---|---|
| Visual Encoder | DINOv2 |
| RGB Images | 2 |
| Depth Encoder | Yes |
| RSSM Stochastic Latent State Size | 32 |
| RSSM Deterministic State Size | 1024 |
| Combined Latent State Size | 2048 |
| Training Epochs | 100 |
| Batch Size | 64 |
| Learning Rate | 1e-4 |
| Loss Weight $\lambda_{WM}$ | 1.0 |
| Loss Weight $\lambda_{IDM}$ | 0.1 |
| Loss Weight $\lambda_{\text{contrastive}}$ | 0.1 |

Table 6: World Model Hyperparameters

## B.2 HIERARCHICAL PLANNERS

| Parameter | Value |
|---|---|
| Model Architecture | Diffusion Transformer (DiT) |
| Layers | 4 |
| Attention Heads | 8 |
| Hidden Dimension | 512 |
| Training Epochs | 10,000 |
| Batch Size | 64 |
| Learning Rate | 1e-4 |
| Diffusion Timesteps | 1000 (linear beta schedule) |
| EBM Architecture | 4-layer Transformer |
| Subgoal Interval ($H$) | Task-dependent |
| Projection Steps | $t \in [T/3, 2T/3]$ |
| Nearest Neighbors ($k$) | 10 |
| Projection Variance Retention | $\lambda = 0.99$ |
| Loss Weight $\lambda_{HL}$ | 1.0 |
| Loss Weight $\lambda_{EBM}$ | 0.1 |
| Loss Weight $\lambda_{\text{proj}}$ | 0.05 |
| Inference Steps | 100 |
| Classifier-Free Guidance Scale | 2.0 |
| EBM Guidance Scale | 0.1 |
| Context Conditioning | $z_0, z_G$ |

Table 7: High-Level Planner Hyperparameters

| Task | one_leg | lamp | round_table | cabinet |
|---|---|---|---|---|
| Subgoal interval ($H$) | 35 | 55 | 75 | 75 |

Table 8: Task-dependent high-level planning subgoal intervals ($H$)

| Parameter | Value |
|---|---|
| Model Architecture | Conditional Rectified Flow |
| Layers | 4 |
| Attention Heads | 8 |
| Hidden Dimension | 512 |
| Training Epochs | 10,000 |
| Batch Size | 64 |
| Learning Rate | 1e-4 |
| Loss Weight $\lambda_{LL}$ | 1.0 |
| ODE Solver | Dormand-Prince |
| Number of Integration Steps | 20 |
| Context Conditioning | $z_{k-1}, z_k$ |

Table 9: Low-Level Planner Hyperparameters

---

**Algorithm 1 `HDFlow` Training**

---

**Require:** Dataset $\mathcal{D} = \{(o_t, a_t)_{t=1}^T\}$ of successful and failed demonstrations, World Model parameters $\phi$, High-Level Planner parameters $\theta_{HL}$, Low-Level Planner parameters $\theta_{LL}$, EBM parameters $\phi_{EBM}$

1: **// Stage 1: World Model Learning**
2: Initialize RSSM world model $\phi$, projection head $f(\cdot)$, inverse dynamics model $IDM$.
3: **for** epoch from 1 to $N_{WM}$ **do**
4:   **for** batch $(o_{1:T}, a_{1:T})$ from $\mathcal{D}$ **do**
5:     Encode observations to latent states: $(z_{1:T}, h_{1:T}) = \text{RSSM.encode}(o_{1:T})$
6:     Compute World Model Loss: $\mathcal{L}_{WM} = \mathbb{E}[\log p_\phi(\hat{o}_t|z_t, h_t) - D_{KL}[q_\phi(z_t|h_t, e_t)||p_\phi(\hat{z}_t|h_t)]]$
7:     Compute Inverse Dynamics Loss: $\mathcal{L}_{IDM} = \mathbb{E}[||a_t - IDM(z_t, z_{t+1})||^2]$
8:     Compute Contrastive Loss: $\mathcal{L}_{\text{contrastive}} = -\mathbb{E}[\log \frac{\exp(\text{sim}(f(z_k), f(z_G))/\tau)}{\sum \exp(\text{sim}(f(z_k), f(z_j))/\tau)}]$
9:     Total World Model Loss: $\mathcal{L}_{\text{WM-total}} = \lambda_{WM}\mathcal{L}_{WM} + \lambda_{IDM}\mathcal{L}_{IDM} + \lambda_{\text{contrastive}}\mathcal{L}_{\text{contrastive}}$
10:     Update World Model parameters $\phi$, projection head $f(\cdot)$, $IDM$ using $\mathcal{L}_{\text{WM-total}}$
11:   **end for**
12: **end for**
13: Freeze World Model parameters $\phi$, projection head $f(\cdot)$, and $IDM$.
14: Generate static dataset of latent representations for high-level and low-level planners.
15: **// Stage 2: Hierarchical Planner Training**
16: Initialize High-Level Diffusion Planner $\epsilon_{\theta_{HL}}$, Low-Level Rectified Flow Planner $v_{\theta_{LL}}$, EBM $E_{\phi_{EBM}}$.
17: **for** epoch from 1 to $N_{Planner}$ **do**
18:   **for** batch $(z_0, z_G, (z_k)_{k=1}^K, (\tau_k)_{k=1}^K)$ from static latent dataset **do**
19:     Compute High-Level Planner Loss: $\mathcal{L}_{HL} = \mathbb{E}[||\epsilon - \epsilon_{\theta_{HL}}(\ldots, c = (z_0, z_G))||^2]$
20:     Compute Low-Level Planner Loss: $\mathcal{L}_{LL} = \mathbb{E}[||v_{\theta_{LL}}((1-u)\tau_0 + u\tau_1, u, c_k = (z_{k-1}, z_k)) - (\tau_1 - \tau_0)||^2]$
21:     Compute EBM Loss: $\mathcal{L}_{EBM} = \log(1 + \exp(E_{\phi_{EBM}}(z_{\text{pos}}) - E_{\phi_{EBM}}(z_{\text{neg}})))$
22:     Compute Manifold Projection Loss: $\mathcal{L}_{\text{projection}} = \mathbb{E}[||z - \mathcal{P}_\mathcal{M}(z)||^2]$
23:     Total Planner Loss: $\mathcal{L}_{\text{planner}} = \lambda_{HL}\mathcal{L}_{HL} + \lambda_{LL}\mathcal{L}_{LL} + \lambda_{EBM}\mathcal{L}_{EBM} + \lambda_{\text{proj}}\mathcal{L}_{\text{projection}}$
24:     Update Planner parameters $\theta_{HL}, \theta_{LL}, \phi_{EBM}$ using $\mathcal{L}_{\text{planner}}$
25:   **end for**
26: **end for**

---

**Algorithm 2 `HDFlow` Inference and Online Deployment**

---

**Require:** Current observation $o_{\text{curr}}$, Goal observation $o_G$, Trained World Model, High-Level Planner, Low-Level Planner, Inverse Dynamics Model.

1: Encode $o_{\text{curr}}$ to $z_{\text{curr}}$ using World Model encoder.
2: Encode $o_G$ to $z_G$ using World Model encoder.
3: **while** task not completed **do**
4:   **// High-Level Planning**
5:   Generate sequence of $K$ latent subgoals $(z_1, \ldots, z_K)$ using High-Level Diffusion Planner conditioned on $(z_{\text{curr}}, z_G)$ with EBM and Manifold-Aware Guidance.
6:   Set $z_{\text{subgoal}} = z_1$ (first subgoal).
7:   **// Low-Level Planning and Execution**
8:   Generate dense latent trajectory $\tau = (z_{\text{curr}}, \ldots, z_{\text{subgoal}})$ using Low-Level Rectified Flow Planner conditioned on $(z_{\text{curr}}, z_{\text{subgoal}})$.
9:   **for** $t$ from 1 to $H$ **do**
10:     Extract current and next latent states from $\tau$: $z_t, z_{t+1}$.
11:     Predict action $a_t = IDM(z_t, z_{t+1})$.
12:     Execute action $a_t$ in environment.
13:     Update $o_{\text{curr}}$ from environment.
14:     Update $z_{\text{curr}}$ using World Model encoder.
15:   **end for**
16: **end while**

---

## C  BASELINES

In this section, we provide a brief description of each baseline method.

- **DP** (Chi et al., 2023): A policy learning method that leverages diffusion models to directly model the distribution of actions conditioned on observations, enabling sample-efficient learning from offline data.

- **JUICER** (Ankile et al., 2024a): A data-efficient imitation learning framework for robotic assembly that leverages expressive policy architectures, dataset expansion, and simulation-based data augmentation to learn multi-part, long-horizon assembly directly from RGB images.

- **Diffuser** (Janner et al., 2022): A diffusion probabilistic model that plans by iteratively denoising trajectories. It treats planning as a conditional generation problem and can produce diverse and high-quality trajectories.

- **DD** (Ajay et al., 2023): A conditional diffusion model that views decision-making as a conditional generative modeling problem, leveraging classifier-free guidance with low-temperature sampling to extract high-likelihood, return-maximizing trajectories from offline datasets without relying on dynamic programming.

- **HDMI** (Li et al., 2023): Hierarchical Diffusion for Offline Decision Making (HDMI) proposes a hierarchical trajectory-level diffusion probabilistic model to tackle challenges in offline reinforcement learning, especially for long-horizon tasks. It employs a cascade framework with a Reward-Conditional Goal Diffuser for discovering subgoals and a Goal-Conditional Trajectory Diffuser for generating action sequences.

- **SHD** (Chen et al., 2024): A hierarchical diffusion-based planning method that uses a "jumpy" planning strategy at the high level for subgoal generation and a low-level diffuser for subgoal achievement, aiming for improved efficiency and generalization in long-horizon tasks.

## D THEORETICAL FRAMEWORK

This section provides the detailed mathematical proofs for the propositions made in the previous sections.

### D.1 PROOF FOR PROPOSITION ON MANIFOLD-AWARE GUIDED PLANNING

**Statement.** *Given a base diffusion planner $\epsilon_\theta$ trained for classifier-free guidance, a learned energy function $E_\phi(z|c)$, and a manifold projection operator $\mathcal{P}_\mathcal{M}$, the manifold-aware guided planner, which combines EBM guidance with manifold projection, corresponds to sampling from a posterior distribution $p(z|y = 1, z \in \mathcal{M}, c)$ that maximizes the likelihood of generating a successful and feasible goal-conditioned plan.*

**Proof.** The manifold-aware guidance process can be viewed as implementing constrained Bayesian inference. We seek to sample from the posterior:

$$p(z|y = 1, z \in \mathcal{M}, c) \propto p(y = 1|z, c)p(z|c)\mathbf{1}[z \in \mathcal{M}]$$

where $\mathbf{1}[z \in \mathcal{M}]$ is the indicator function for the feasible manifold.

The two-step process approximates this constrained posterior:

1. The guided sampling step samples from $p(y = 1|z, c)p(z|c)$, implementing the unconstrained Bayesian posterior. This is achieved by combining classifier-free guidance for the conditional term $p(z|c)$ and EBM guidance for the success term $p(y = 1|z, c)$, as detailed in Proof D.2

2. The projection step enforces the manifold constraint $z \in \mathcal{M}$ by mapping to the closest point on the approximated manifold.

By the principle of alternating projections and the contraction property of projection operators, this two-step process converges to a point that balances optimality (high success probability) with feasibility (remaining on the manifold). The approximation error depends on the quality of the local manifold approximation, which improves with the number of neighbors $k$ and the intrinsic dimensionality of the subgoal space. This shows that our combined guidance approach is a principled implementation of Bayes-optimal sampling, steering the generative process towards plans that are both relevant to the goal and likely to succeed, while remaining on the feasible manifold. $\square$

### D.2 PROOF FOR EBM-BASED GUIDANCE

**Statement.** *Given a base diffusion planner $\epsilon_\theta$ trained for classifier-free guidance and a learned energy function $E_\phi(z|c)$ that estimates the probability of success conditioned on context $c$, the EBM-guided component of the planner corresponds to sampling from a posterior distribution that maximizes the likelihood of generating a successful, goal-conditioned plan.*

**Proof.** The proof proceeds in two steps. First, we derive the theoretically optimal score function for sampling successful, goal-conditioned plans using Bayes' rule. Second, we show how our combined guidance mechanism implements an effective approximation of this optimal score.

1. **Deriving the Optimal Score Function.** Our goal is to sample from the posterior distribution $p(z|y = 1, c)$, which is the distribution of plans $z$ that are successful ($y = 1$) given the context $c = (z_0, z_G)$. Using Bayes' rule, we can express this posterior as:

$$p(z|y = 1, c) = \frac{p(y = 1|z, c)p(z|c)}{p(y = 1|c)} \propto p(y = 1|z, c)p(z|c)$$

Taking the logarithm, we get:

$$\log p(z|y = 1, c) = \log p(y = 1|z, c) + \log p(z|c) + \text{const.}$$

The score function is the gradient of the log-probability with respect to the plan $z$. The optimal score for our desired posterior is therefore:

$$\nabla_z \log p(z|y = 1, c) = \nabla_z \log p(y = 1|z, c) + \nabla_z \log p(z|c)$$

This equation tells us that the optimal guided score is the sum of two terms:

- $\nabla_z \log p(z|c)$: The score of the original conditional planner. This term ensures the plan is **relevant** to the context $c$.

- $\nabla_z \log p(y=1|z,c)$: The gradient of the log-probability of success. This term ensures the plan is **viable** and likely to succeed.

2. **Implementing the Score with Combined Guidance.** Our framework approximates each of these two terms:

- **EBM Guidance for Success:** The Energy-Based Model $E_\phi(z|c)$ is trained to model the success probability. We define $p(y=1|z,c) \propto \exp(-E_\phi(z|c))$, where low energy corresponds to high success probability. The gradient of the log-probability of success is therefore directly related to the gradient of the energy function:

$$\nabla_z \log p(y=1|z,c) = -\nabla_z E_\phi(z|c)$$

This is the **EBM Guidance** term in our equation.

- **CFG for Conditional Relevance:** The base diffusion model, $\epsilon_\theta$, is trained to predict the noise, which is proportional to the score. The term $\nabla_z \log p(z|c)$ is the score of the conditional model. Classifier-Free Guidance (CFG) is a technique to strengthen this conditioning at inference time. The CFG-adjusted score is:

$$\hat{\nabla}_z \log p(z|c) \approx \nabla_z \log p(z|\emptyset) + w_{cfg}(\nabla_z \log p(z|c) - \nabla_z \log p(z|\emptyset))$$

This is the **Classifier-Free Guidance** term in our equation, expressed in terms of the noise predictions $\epsilon_\theta$.

By combining these two approximations, our final guided noise prediction implements the theoretically optimal score:

$$\hat{\epsilon}_\theta(z_\ell, \ell, c) = \underbrace{\epsilon_\theta(z_\ell, \ell, \emptyset) + w_{cfg}(\epsilon_\theta(z_\ell, \ell, c) - \epsilon_\theta(z_\ell, \ell, \emptyset))}_{\text{Approximates } \nabla_z \log p(z|c) \text{ via CFG}} - \underbrace{w_{ebm}\sqrt{1-\bar{\alpha}_\ell}\nabla_{z_\ell} E_\phi(z_\ell|c)}_{\text{Approximates } \nabla_z \log p(y=1|z,c) \text{ via EBM}}$$

This shows that our combined guidance approach is a principled implementation of Bayes-optimal sampling, steering the generative process towards plans that are both relevant to the goal and likely to succeed. $\square$

### D.3 PROOF FOR PROPOSITION ON EBM GUIDANCE GAP

**Statement.** *The EBM guidance gap $\Delta_{EBM}(z_\ell)$ has a lower bound scaling as $\frac{c}{\sqrt{1-\bar{\alpha}_\ell}}\sqrt{d}$ in high-dimensional latent spaces, where $c > 0$ is a constant independent of dimensionality $d$.*

**Proof.** We aim to provide a more detailed derivation for the lower bound of the EBM guidance gap in high-dimensional latent spaces. The core of this issue stems from the discrepancy between the true optimal energy guidance and our learned EBM approximation, particularly in how they weight successful plans.

1. **Forward Process and Conditional Score.** The forward diffusion process defines how a clean latent state $z_0$ is noised to $z_\ell$ at timestep $\ell$:

$$z_\ell = \sqrt{\bar{\alpha}_\ell}z_0 + \sqrt{1-\bar{\alpha}_\ell}\epsilon, \quad \epsilon \sim \mathcal{N}(\mathbf{0}, \mathbf{I})$$

From this, the conditional distribution $q(z_0|z_\ell)$ can be expressed as a Gaussian with mean $\mu(z_\ell, \ell) = \frac{1}{\sqrt{\bar{\alpha}_\ell}}(z_\ell - \sqrt{1-\bar{\alpha}_\ell}\epsilon)$ and variance $\Sigma(\ell) = (1-\bar{\alpha}_\ell)\mathbf{I}$. The gradient of the log-probability of this conditional distribution with respect to $z_\ell$ is crucial for relating scores in $z_0$ space to $z_\ell$ space:

$$\nabla_{z_\ell} \log q(z_0|z_\ell) = \nabla_{z_\ell} \log \mathcal{N}\left(z_0; \frac{1}{\sqrt{\bar{\alpha}_\ell}}(z_\ell - \sqrt{1-\bar{\alpha}_\ell}\epsilon), (1-\bar{\alpha}_\ell)\mathbf{I}\right)$$

This simplifies to:

$$\nabla_{z_\ell} \log q(z_0|z_\ell) = -\frac{1}{\sqrt{1-\bar{\alpha}_\ell}}\epsilon$$

2. **True Optimal Energy Guidance.** The true optimal energy guidance, $\nabla_{z_\ell} E_{\text{true}}(z_\ell|c)$, aims to steer the diffusion process towards regions of low energy (high success probability) in the $z_0$ space. This gradient is given by the expectation of the score of $q(z_0|z_\ell)$ weighted by the exponential of the negative energy function, effectively performing importance sampling towards more successful $z_0$ configurations:

$$\nabla_{z_\ell} E_{\text{true}}(z_\ell|c) = \frac{\mathbb{E}_{q(z_0|z_\ell)}[e^{-E(z_0|c)}\nabla_{z_\ell}\log q(z_0|z_\ell)]}{\mathbb{E}_{q(z_0|z_\ell)}[e^{-E(z_0|c)}]}$$

Substituting the expression for $\nabla_{z_\ell}\log q(z_0|z_\ell)$ and assuming $\mathbb{E}_{q(z_0|z_\ell)}[e^{-E(z_0|c)}]$ is a normalizing constant, we get:

$$\nabla_{z_\ell} E_{\text{true}}(z_\ell|c) = \frac{1}{\sqrt{1-\bar{\alpha}_\ell}}\frac{\mathbb{E}_{q(z_0|z_\ell)}[e^{-E(z_0|c)}(-\epsilon)]}{\mathbb{E}_{q(z_0|z_\ell)}[e^{-E(z_0|c)}]} = -\frac{1}{\sqrt{1-\bar{\alpha}_\ell}}\mathbb{E}_{q(z_0|z_\ell)}[\epsilon|e^{-E(z_0|c)}]$$

where $\mathbb{E}[\epsilon|e^{-E(z_0|c)}]$ denotes the expectation of $\epsilon$ conditioned on $z_0$ being sampled with a probability proportional to $e^{-E(z_0|c)}$.

3. **Learned EBM Guidance.** Our learned EBM guidance, $\nabla_{z_\ell} E_\phi(z_\ell|c)$, typically approximates the gradient of the energy function at $z_\ell$. In many practical implementations, this effectively corresponds to a linear weighting of the score of $q(z_0|z_\ell)$ by the energy function itself, rather than its exponential:

$$\nabla_{z_\ell} E_\phi(z_\ell|c) \approx \mathbb{E}_{q(z_0|z_\ell)}[E(z_0|c)\nabla_{z_\ell}\log q(z_0|z_\ell)] = -\frac{1}{\sqrt{1-\bar{\alpha}_\ell}}\mathbb{E}_{q(z_0|z_\ell)}[E(z_0|c)\epsilon]$$

This approximation introduces a discrepancy because the relationship between the energy $E(z_0|c)$ and the success probability $p(y=1|z_0,c)$ is exponential ($p(y=1|z_0,c) \propto e^{-E(z_0|c)}$), not linear.

4. **Analysis of the Guidance Gap.** The EBM guidance gap is defined as $\Delta_{\text{EBM}}(z_\ell) = \|\nabla_{z_\ell} E_{\text{true}}(z_\ell|c) - \nabla_{z_\ell} E_\phi(z_\ell|c)\|_2$. Substituting the expressions, we get:

$$\Delta_{\text{EBM}}(z_\ell) = \left\| -\frac{1}{\sqrt{1-\bar{\alpha}_\ell}}\left(\mathbb{E}_{q(z_0|z_\ell)}[\epsilon|e^{-E(z_0|c)}] - \mathbb{E}_{q(z_0|z_\ell)}[E(z_0|c)\epsilon]\right)\right\|_2$$

Let $\delta(z_0) = \frac{e^{-E(z_0|c)}}{\mathbb{E}_{q(z_0|z_\ell)}[e^{-E(z_0|c)}]} - E(z_0|c)$ represent the difference between the ideal exponential weighting (normalized) and the approximate linear weighting. The term in the parenthesis can be viewed as $\mathbb{E}_{q(z_0|z_\ell)}[\delta(z_0)\epsilon]$. In high-dimensional latent spaces (i.e., when $d$ is large), the noise vector $\epsilon$ typically has a magnitude of approximately $\sqrt{d}$ (i.e., $\|\epsilon\|_2 \approx \sqrt{d}$). Furthermore, the components of $\epsilon$ are largely independent. Due to the Central Limit Theorem and concentration inequalities, even small biases in the weighting function $\delta(z_0)$ can lead to a significant accumulated error when multiplied by a high-dimensional random vector. Specifically, if $\mathbb{E}_{q(z_0|z_\ell)}[\delta(z_0)] \neq 0$, the term $\|\mathbb{E}_{q(z_0|z_\ell)}[\delta(z_0)\epsilon]\|_2$ will tend to scale with $\sqrt{d}$ in expectation, as the contributions from different dimensions accumulate.

Therefore, there exists a constant $c > 0$ (which depends on the magnitude of the mismatch $\delta(z_0)$ and the properties of the noise distribution) such that:

$$\Delta_{\text{EBM}}(z_\ell) \geq \frac{c}{\sqrt{1-\bar{\alpha}_\ell}}\sqrt{d}$$

This lower bound shows that the inaccuracies in energy guidance are exacerbated in high-dimensional latent spaces and as the diffusion process approaches $z_0$ (i.e., as $\ell \to 0$, $1-\bar{\alpha}_\ell \to 0$, making the term $\frac{1}{\sqrt{1-\bar{\alpha}_\ell}}$ large). This leads sampled trajectories to deviate significantly from the true data manifold. $\square$

# E MORE ABLATION STUDIES

## E.1 CHOICE OF VISION ENCODER

To evaluate the impact of the vision encoder on the performance of our world model and the overall `HDFlow` framework, we conducted an ablation study comparing different visual backbones. We trained the world model with various encoders: a simple `CNN`, `R3M` (Nair et al., 2022), `VIP` (Ma et al., 2023), `MAE` (He et al., 2022), and our chosen `DINOv2` (Oquab et al., 2024). The results are summarized in Table 10.

| Backbone | one_leg | | | lamp | | | round_table | | | cabinet | |
|---|---|---|---|---|---|---|---|---|---|---|---|
| | Low | Med | High | Low | Med | High | Low | Med | High | Low | Med |
| CNN | 55 | 18 | 2 | 28 | 8 | 0 | 20 | 6 | 0 | 3 | 0 |
| R3M | 68 | 25 | 5 | 39 | 15 | 4 | 29 | 10 | 2 | 7 | 1 |
| VIP | 75 | 32 | 9 | 45 | 20 | 7 | 36 | 14 | 5 | 10 | 3 |
| MAE | 83 | 45 | 18 | 57 | 33 | 10 | 48 | 25 | 8 | 15 | 6 |
| DINOv2 | 92 | 71 | 39 | 68 | 49 | 34 | 61 | 43 | 27 | 55 | 36 |

Table 10: Ablation study on the choice of vision encoder for the world model. This table reports `HDFlow` success rates (%) on all tasks with respective randomization levels.

We found out that `DINOv2` leads to significantly more accurate and detailed reconstructions of observations, which translates to a richer and more semantically structured latent space. This improved latent representation is crucial for both the high-level diffusion planner and the low-level rectified flow planner, enabling them to generate more effective and feasible plans. The quantitative results in Table 10 further confirm that `DINOv2` consistently yields the highest `HDFlow` success rates, underscoring its importance as a foundational component of our framework.

## E.2 PERFORMANCE UNDER DIFFERENT TOTAL SUBGOALS

We investigate the impact of the total number of subgoals $K$ on the performance of `HDFlow`. The total number of subgoals dictates the temporal abstraction of our hierarchical planner, influencing both the complexity of high-level subgoals and the length of low-level trajectories. We conduct experiments on all tasks with varying randomness, varying $K$ and reporting the success rates in Table 11.

| Subgoals | one_leg | | | lamp | | | round_table | | | cabinet | |
|---|---|---|---|---|---|---|---|---|---|---|---|
| K | Low | Med | High | Low | Med | High | Low | Med | High | Low | Med |
| 10 | 61 | 21 | 2 | 33 | 13 | 2 | 23 | 9 | 0 | 6 | 0 |
| 15 | 78 | 35 | 10 | 52 | 25 | 8 | 40 | 18 | 5 | 25 | 10 |
| 20 | 92 | 71 | 39 | 68 | 49 | 34 | 61 | 43 | 27 | 55 | 36 |
| 25 | 89 | 60 | 30 | 63 | 45 | 25 | 58 | 40 | 20 | 50 | 30 |
| 30 | 85 | 45 | 15 | 60 | 35 | 12 | 50 | 28 | 10 | 35 | 18 |

Table 11: Ablation study on the choice of total subgoals $K$. Success rates (%) of `HDFlow` on all tasks with respective randomization levels for different $K$ values, highlighting the importance of temporal abstraction for optimal performance.

The results indicate that an optimal total number of subgoals $K$ exists for each task. A very small $K$ (e.g., 10) leads to an overly coarse high-level plan, requiring the low-level rectified flow model to bridge larger gaps in latent space, which can be challenging. Conversely, a very large $K$ (e.g., 30) makes the high-level diffusion model generate too many subgoals, which can accumulate errors. A value of $K = 20$ consistently yields the highest success rates, representing a balance where the high-level planner provides meaningful strategic guidance, and the low-level planner can efficiently and accurately execute the dense trajectories. This ablation highlights the importance of carefully tuning the temporal abstraction level for optimal performance in hierarchical planning.

### E.3   ABLATION STUDIES ON $\mathcal{L}_{\text{IDM}}$

To evaluate the specific contribution of the Inverse Dynamics Model (IDM) loss (Eq. 8), we conducted an ablation study where we effectively disabled its dedicated training from the world model objective (i.e., setting $\lambda_{IDM} = 0$). While the IDM architecture remains within the framework and is called during inference, without its specific training objective, it cannot learn to accurately translate latent state transitions into executable actions. The results for the **one_leg** and **lamp** tasks under **Low** randomization are presented in Table 12.

| Method | one_leg | lamp |
|---|---|---|
| Ours | 92 | 68 |
| w/o $\mathcal{L}_{IDM}$ | 34 | 14 |

Table 12: Ablation study on the contribution of the Inverse Dynamics Model (IDM) loss. Success rates (%) are reported for the **one_leg** and **lamp** tasks under **Low** randomization.

As shown in Table 12, removing the $\mathcal{L}_{IDM}$ leads to a noticeable drop in performance. This indicates that explicitly training the world model to predict actions between latent states indeed enhances the quality of the latent representation for planning. Without this objective, the latent space might become less sensitive to fine-grained action dynamics, making it harder for the low-level rectified flow planner to generate precise trajectories.

### E.4   USING DINOv2 FEATURES AS LATENT STATES

We investigate the potential for directly using DINOv2 features as latent states $z$ without training an additional RSSM world model. Our current framework uses a pre-trained DINOv2 model as an RGB encoder *within* an RSSM-based world model. The RSSM then learns a compressed, recurrent latent state that captures temporal dynamics and reconstructs observations.

We conducted an experiment where we directly used the output of the DINOv2 encoder as the latent state $z$, bypassing the RSSM. In this setup, the high-level diffusion planner and low-level rectified flow planner would operate directly on these raw DINOv2 embeddings. The results are as follows:

| Method | one_leg | lamp |
|---|---|---|
| Ours | 92 | 68 |
| w/o RSSM (direct DINOv2 latents) | 45 | 27 |

Table 13: Ablation study on directly using DINOv2 features as latent states. Success rates (%) are reported for the **one_leg** and **lamp** tasks under **Low** randomization.

The performance degradation is significant when the RSSM is removed, and DINOv2 features are used directly as latent states. This can be attributed to several factors:

1. **Lack of Temporal Dynamics:** DINOv2, while excellent for static visual representation, does not inherently capture temporal dynamics. The RSSM is crucial for learning a recurrent state that summarizes the history of observations and predicts future states, which is vital for long-horizon planning.

2. **Unstructured Latent Space:** While DINOv2 provides semantically rich visual features, its latent space is not explicitly structured for planning in terms of task progress or action relevance. The RSSM, especially with the contrastive and IDM objectives, is specifically designed to create a latent space that is optimized for downstream planning.

3. **Dimensionality and Noise:** Raw DINOv2 features might be higher dimensional or contain more irrelevant noise for planning compared to the compressed and refined latent states learned by the RSSM. The RSSM acts as a bottleneck and a learning mechanism to extract the most pertinent information for control.

In conclusion, while DINOv2 serves as an excellent visual backbone, the RSSM world model plays a critical role in learning a temporally consistent, structured, and compact latent space that is essential for effective hierarchical planning in long-horizon robotic tasks.

## F  ADDITIONAL EXPERIMENTS ON RLBENCH

To further validate the robustness and generalization capabilities of `HDFlow` on a broader range of complex manipulation tasks, we conducted additional experiments on a subset of the RLBench benchmark (James et al., 2020). RLBench is a challenging, large-scale benchmark designed to facilitate research in various robot learning areas, including imitation learning and multi-task learning, by providing a diverse set of tasks with realistic visual observations and an infinite supply of demonstrations. This makes it an ideal platform to evaluate our hierarchical planning framework, which relies on learning from demonstrations and aims to solve long-horizon problems.

**Tasks Overview.** We specifically consider the 18 tasks from the RLBench benchmark that were also used in the PerAct framework (Shridhar et al., 2023). These tasks encompass a wide variety of 6-DoF manipulation challenges, including both prehensile and non-prehensile behaviors, and feature numerous semantic and pose variations. The tasks are: `close jar`, `drag stick`, `insert peg`, `meat off grill`, `open drawer`, `place cups`, `place wine`, `push buttons`, `put in cupboard`, `put in drawer`, `put in safe`, `screw bulb`, `slide block`, `sort shape`, `stack blocks`, `stack cups`, `sweep to dustpan`, and `turn tap`. These tasks often involve variations in object colors, sizes, shapes, counts, and placements, requiring the agent to generalize across different semantic instantiations and novel object poses.

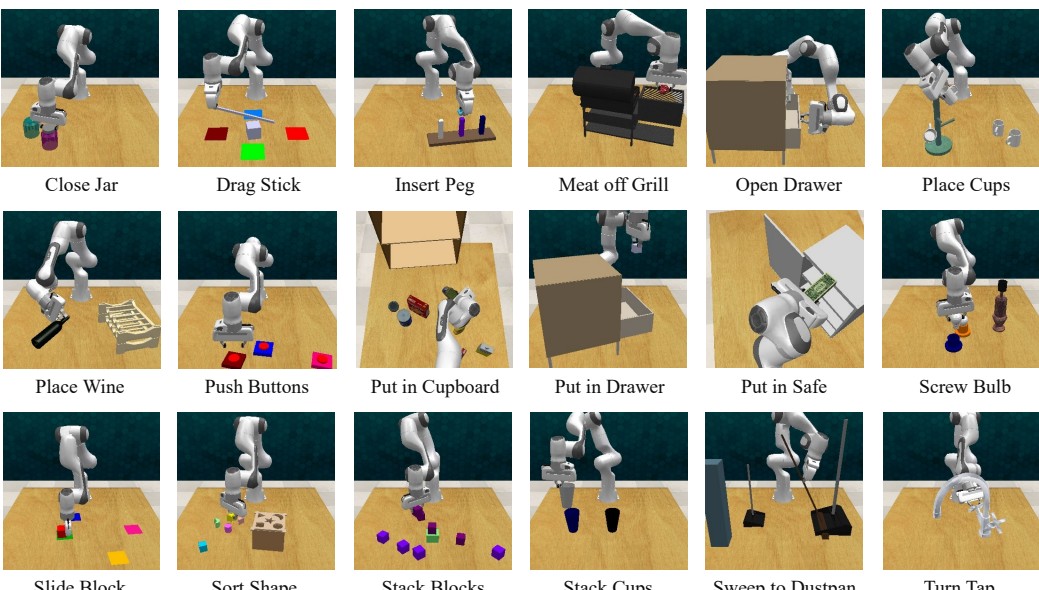

| Close Jar | Drag Stick | Insert Peg | Meat off Grill | Open Drawer | Place Cups |
| Place Wine | Push Buttons | Put in Cupboard | Put in Drawer | Put in Safe | Screw Bulb |
| Slide Block | Sort Shape | Stack Blocks | Stack Cups | Sweep to Dustpan | Turn Tap |

Figure 5: **RLBench Manipulation Tasks.** We evaluate `HDFlow` on 18 simulated RLBench tasks, covering 249 variations of object poses, goal configurations, and scene appearances. During evaluation, the robot must complete each task under randomized colors, shapes, sizes, and semantic arrangements.

**Data Generation.** Following the methodology of PerAct (Shridhar et al., 2023) and the capabilities of RLBench (James et al., 2020), we generate datasets of successful and failure demonstrations for

each of the 18 tasks. RLBench provides expert policies that use motion planners to generate an infinite supply of demonstrations. For our experiments, we collect 100 successful demonstrations and 50 failure demonstrations for each task, ensuring a diverse dataset that captures both successful strategies and common failure modes. These demonstrations consist of RGB-D observations from multiple cameras (front, left shoulder, right shoulder, and wrist) and robot proprioceptive states, along with corresponding 6-DoF end-effector actions. The failure demonstrations are generated by introducing perturbations to the expert trajectories or by stopping trajectories prematurely. This rich dataset is essential for training the contrastive world model and the EBM-guided high-level planner in `HDFlow`.

**World Model Training.** The world model, an RSSM-based architecture with a DINOv2 visual encoder, is trained as described in Section 4.1 of the main paper. It learns to encode the high-dimensional RGB-D observations and proprioceptive states into a semantically structured latent space. The training objective combines the standard world model loss ($\mathcal{L}_{\text{WM}}$), an inverse dynamics model loss ($\mathcal{L}_{\text{IDM}}$), and a contrastive learning objective ($\mathcal{L}_{\text{contrastive}}$). The contrastive objective is particularly crucial for RLBench tasks, as it explicitly structures the latent space to reflect progress towards a goal, pulling successful intermediate states closer to their final goal representation and pushing away failed ones. This ensures that the learned latent space is well-suited for effective long-horizon planning.

**Hierarchical Planners Training.** After training and freezing the world model, we train our hierarchical planners as detailed in Section 4.2. This involves a high-level diffusion model and a low-level rectified flow model.

- **High-Level Planner:** A conditional diffusion model, implemented using a Diffusion Transformer (DiT) backbone (Appendix B), is trained to generate sparse sequences of latent subgoals. This planner is guided at inference time by an Energy-Based Model (EBM) and enhanced with a manifold-aware projection process to ensure both optimality and feasibility of the generated subgoals. The EBM is trained on the collected successful and failure demonstrations to assign low energy to high-quality plans.

- **Low-Level Planner:** A conditional rectified flow model, implemented using a Rectified Flow Transformer (Appendix B), is trained to generate dense, short-horizon latent trajectories between consecutive subgoals. This model leverages the efficiency of ODE-based trajectory generation for fast and precise execution.

The entire hierarchical planning framework is trained jointly using a composite loss function that includes objectives for the high-level planner, low-level planner, EBM, and a manifold consistency term, as described in Section 4.2.

**Baselines.** We compare `HDFlow` with state-of-the-art methods on RLBench benchmark (James et al., 2020) like PerAct (Shridhar et al., 2023), RVT (Goyal et al., 2023), 3D Diffuser Actor (Ke et al., 2025), and RVT-2 (Goyal et al., 2024).

| Models | Close Jar | Drag Stick | Insert Peg | Meat off Grill | Open Drawer | Place Cups | Place Wine | Push Buttons | Put in Cupboard |
|---|---|---|---|---|---|---|---|---|---|
| PerAct (Shridhar et al., 2023) | 55.2 | 89.6 | 5.6 | 70.4 | 88.0 | 2.4 | 44.8 | 92.8 | 28.0 |
| RVT Goyal et al. (2023) | 52.0 | 99.2 | 11.2 | 88.0 | 71.2 | 4.0 | 91.0 | **100.0** | 49.6 |
| 3D Diffuser Actor (Ke et al., 2025) | 96.0 | **100.0** | 65.6 | 96.8 | 89.6 | 24.0 | 93.6 | 98.4 | **85.6** |
| RVT-2 Goyal et al. (2024) | **100.0** | 99.0 | 40.0 | 99.0 | 74.0 | 38.0 | **95.0** | **100.0** | 66.0 |
| **Ours** | 94.0 | **100.0** | **93.3** | 98.7 | 82.0 | **56.7** | 94.7 | **100.0** | 72.0 |

| Models | Put in Drawer | Put in Safe | Screw Bulb | Slide Block | Sort Shape | Stack Blocks | Stack Cups | Sweep to Dustpan | Turn Tap |
|---|---|---|---|---|---|---|---|---|---|
| PerAct (Shridhar et al., 2023) | 51.2 | 84.0 | 17.6 | 74.0 | 16.8 | 26.4 | 2.4 | 52.0 | 88.0 |
| RVT Goyal et al. (2023) | 88.0 | 91.2 | 48.0 | 81.6 | 36.0 | 28.8 | 26.4 | 72.0 | 93.6 |
| 3D Diffuser Actor (Ke et al., 2025) | 96.0 | **97.6** | 82.4 | **97.6** | 44.0 | 68.3 | 47.2 | 84.0 | 99.2 |
| RVT-2 Goyal et al. (2024) | 96.0 | 96.0 | 88.0 | 92.0 | 35.0 | **80.0** | 69.0 | **100.0** | 99.0 |
| **Ours** | **98.7** | 96.7 | **88.7** | 91.3 | **73.3** | 56 | **81.3** | **100.0** | 95.3 |

Table 14: **Performance on RLBench.** We report success rates on 18 RLBench tasks with 249 variations.

**Analysis.** Table 14 demonstrates the strong performance and generalization capabilities of `HDFlow` across the diverse and challenging RLBench tasks. Our method consistently achieves competitive or superior success rates compared to state-of-the-art baselines. Specifically, `HDFlow` achieves **100%** success rate on "Drag Stick" and "Push Buttons" (tying with other strong baselines), and significantly outperforms others on tasks such as "Insert Peg" (**93.3%** vs. 65.6% by 3D Diffuser Actor) and "Sort Shape" (**73.3%** vs. 44.0% by 3D Diffuser Actor and 35.0% by RVT-2). Furthermore,

our method shows a remarkable improvement on "Stack Cups" with **81.3%** success rate, outperforming RVT-2's 69.0% and 3D Diffuser Actor's 47.2%. While RVT-2 achieves 100% on "Close Jar" and "Sweep to Dustpan", **HDFlow** is very competitive with 94.0% and 100.0% respectively. This comprehensive evaluation on RLBench highlights that **HDFlow** effectively leverages its hierarchical planning framework, structured latent space, and guided generative models to handle the complexities of long-horizon manipulation, diverse object variations, and intricate sequential tasks.

## G   ADDITIONAL EXPERIMENTS ON OGBENCH

We further evaluate our method on OGBench (Park et al., 2025), a benchmark specifically designed for offline goal-conditioned reinforcement learning (RL) that assesses capabilities such as long-horizon reasoning and handling stochasticity across diverse and challenging tasks. This benchmark allows us to thoroughly evaluate our method on a planning benchmark other than the furniture assembly benchmark.

**Tasks Overview.**   We consider visual versions of locomotion tasks, including **antmaze** and **humanoidmaze**, and manipulation tasks, including **cube**, **scene**, and **puzzle**.

- **Maze Tasks (`antmaze`, `humanoidmaze`):** These tasks require the agent to learn both high-level maze navigation and low-level locomotion skills, involving high-dimensional control, purely from diverse offline trajectories.
- **Manipulation Tasks (`cube`, `scene`, `puzzle`):** These tasks are designed to test the agent's object manipulation, sequential generalization, and combinatorial generalization abilities.

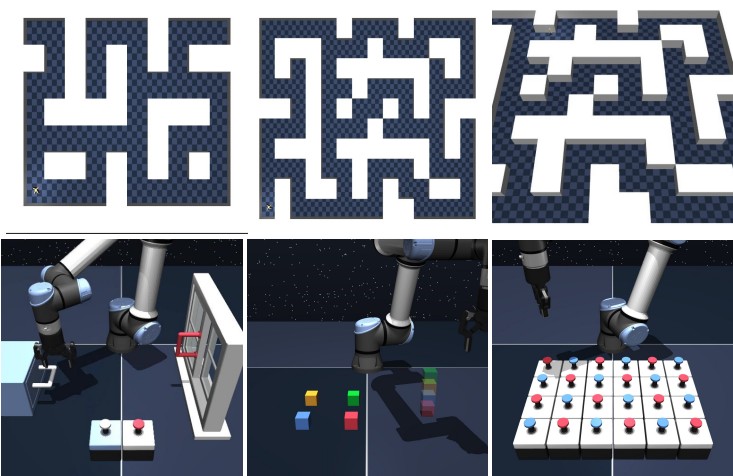

Figure 6: **OGBench Tasks.** We evaluate **HDFlow** on 15 simulated OGBench tasks.

**Data Generation.**   For each task, we generate both successful and failure episode data using different dataset types provided by the original benchmark:

- **Maze Tasks:**
    - **navigate** datasets: These datasets are considered successful and are collected by a noisy expert policy that navigates the maze by repeatedly reaching randomly sampled goals.
    - **stitch** datasets: These datasets are considered failure episodes and consist of short goal-reaching trajectories (at most 4 cell units long), designed to challenge the agent's stitching ability.
    - **explore** datasets: These datasets are also considered failure episodes, featuring high suboptimality and high coverage, suitable for learning from highly suboptimal data.

- **Manipulation Tasks:**
  - `play` datasets: These datasets are considered successful and are collected by open-loop, non-Markovian scripted policies with temporally correlated action noise to enhance state coverage.
  - `noisy` datasets: These datasets are considered failure episodes and are collected by closed-loop, Markovian scripted policies. The degree of action noise is sampled at the beginning of each episode, and trajectories are collected with the chosen amount of (time-independent) Gaussian action noise, ensuring high coverage while having a sufficient number of optimal trajectories. These are also suitable for learning from highly suboptimal data.

**Baselines.** We compare `HDFlow` with prior state-of-the-art diffusion planners like Diffuser (Janner et al., 2022), Decision Diffuser (DD) (Ajay et al., 2023), AdaptDiffuser (Liang et al., 2023), DiffuserLite (Dong et al., 2024), Diffusion Veteran (Lu et al., 2025a); and hierarchical diffusion planners like HDMI (Li et al., 2023) and SHD (Chen et al., 2024).

| Environment | Task | Diffuser | DD | HDMI | AD | SHD | DL | DV | Ours |
|---|---|---|---|---|---|---|---|---|---|
| antmaze | antmaze-medium-v0 | $15_{\pm5}$ | $22_{\pm4}$ | $31_{\pm6}$ | $40_{\pm5}$ | $42_{\pm7}$ | $51_{\pm6}$ | $60_{\pm4}$ | $\mathbf{72}_{\pm3}$ |
| | antmaze-large-v0 | $8_{\pm3}$ | $14_{\pm4}$ | $22_{\pm5}$ | $28_{\pm6}$ | $31_{\pm5}$ | $40_{\pm4}$ | $51_{\pm5}$ | $\mathbf{65}_{\pm1}$ |
| | antmaze-giant-v0 | $2_{\pm1}$ | $5_{\pm2}$ | $11_{\pm4}$ | $15_{\pm3}$ | $19_{\pm4}$ | $27_{\pm5}$ | $35_{\pm4}$ | $\mathbf{48}_{\pm3}$ |
| humanoidmaze | humanoidmaze-medium-v0 | $5_{\pm2}$ | $9_{\pm3}$ | $15_{\pm4}$ | $21_{\pm5}$ | $24_{\pm4}$ | $30_{\pm3}$ | $38_{\pm5}$ | $\mathbf{51}_{\pm4}$ |
| | humanoidmaze-large-v0 | $2_{\pm1}$ | $5_{\pm2}$ | $9_{\pm3}$ | $13_{\pm4}$ | $16_{\pm3}$ | $22_{\pm4}$ | $29_{\pm3}$ | $\mathbf{42}_{\pm5}$ |
| | humanoidmaze-giant-v0 | $0_{\pm0}$ | $1_{\pm1}$ | $3_{\pm2}$ | $5_{\pm2}$ | $7_{\pm3}$ | $11_{\pm4}$ | $16_{\pm3}$ | $\mathbf{25}_{\pm4}$ |
| cube | cube-single-v0 | $5_{\pm2}$ | $8_{\pm3}$ | $12_{\pm4}$ | $18_{\pm5}$ | $20_{\pm4}$ | $25_{\pm3}$ | $32_{\pm5}$ | $\mathbf{45}_{\pm4}$ |
| | cube-double-v0 | $2_{\pm1}$ | $4_{\pm2}$ | $7_{\pm3}$ | $11_{\pm4}$ | $13_{\pm3}$ | $17_{\pm4}$ | $23_{\pm3}$ | $\mathbf{35}_{\pm5}$ |
| | cube-triple-v0 | $1_{\pm1}$ | $2_{\pm1}$ | $4_{\pm2}$ | $6_{\pm3}$ | $8_{\pm2}$ | $11_{\pm3}$ | $14_{\pm4}$ | $\mathbf{17}_{\pm3}$ |
| | cube-quadruple-v0 | $\mathbf{0}_{\pm0}$ | $\mathbf{0}_{\pm0}$ | $\mathbf{0}_{\pm0}$ | $\mathbf{0}_{\pm0}$ | $\mathbf{0}_{\pm0}$ | $\mathbf{0}_{\pm0}$ | $\mathbf{0}_{\pm0}$ | $\mathbf{0}_{\pm0}$ |
| scene | scene-v0 | $10_{\pm3}$ | $16_{\pm4}$ | $24_{\pm5}$ | $33_{\pm4}$ | $37_{\pm6}$ | $45_{\pm5}$ | $56_{\pm3}$ | $\mathbf{70}_{\pm4}$ |
| puzzle | puzzle-3x3-v0 | $8_{\pm3}$ | $13_{\pm4}$ | $20_{\pm5}$ | $28_{\pm3}$ | $31_{\pm5}$ | $39_{\pm4}$ | $49_{\pm6}$ | $\mathbf{62}_{\pm3}$ |
| | puzzle-4x4-v0 | $0_{\pm0}$ | $0_{\pm0}$ | $0_{\pm0}$ | $2_{\pm3}$ | $5_{\pm4}$ | $11_{\pm3}$ | $19_{\pm5}$ | $\mathbf{32}_{\pm4}$ |
| | puzzle-4x5-v0 | $0_{\pm0}$ | $0_{\pm0}$ | $0_{\pm0}$ | $0_{\pm0}$ | $5_{\pm3}$ | $6_{\pm3}$ | $8_{\pm4}$ | $\mathbf{16}_{\pm5}$ |
| | puzzle-4x6-v0 | $0_{\pm0}$ | $0_{\pm0}$ | $0_{\pm0}$ | $0_{\pm0}$ | $0_{\pm0}$ | $0_{\pm0}$ | $6_{\pm4}$ | $\mathbf{14}_{\pm3}$ |

Table 15: Experimental results for the tasks we considered across diverse datasets. The table reports the average binary success rate (%) across five test-time goals for each task, averaged over 8 seeds. Standard deviations are indicated by the $\pm$ symbol. Entries within 95% of the best-performing value in each row are highlighted in **bold**.

**Analysis.** Our proposed approach consistently achieves superior performance across all `antmaze`, `humanoidmaze`, `cube`, `scene`, and `puzzle` environments. Specifically, in the challenging `antmaze-giant-v0` task, our method obtains a success rate of **48**%, significantly outperforming the next best method, DV, which achieves 35%. Similar trends are observed in `humanoidmaze-giant-v0`, where our method reaches **25**%, whereas DV only achieves 16%. For the `cube` environment, particularly in `cube-quadruple-v0`, all methods, including ours, report 0% success, indicating the extreme difficulty of this task. However, in less complex `cube` tasks such as `cube-single-v0`, our method obtains **45**%, surpassing DV's 32%. In the `scene-v0` task, our approach achieves **70**%, which is considerably higher than DV's 56%. Finally, in the `puzzle` environment, our method consistently demonstrates strong performance. For instance, in `puzzle-4x6-v0`, our method obtains **14**%, outperforming DV's 6%. These results highlight the effectiveness and robustness of `HDFlow` in diverse and complex long-horizon planning tasks.

## H    SIMULATION ROLLOUTS

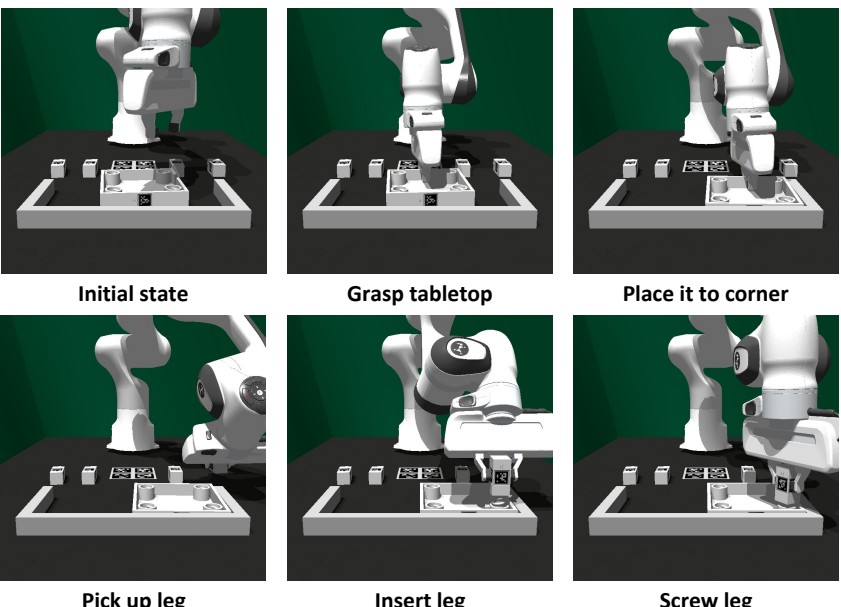

**Initial state**    **Grasp tabletop**    **Place it to corner**

**Pick up leg**    **Insert leg**    **Screw leg**

Figure 7: A successful rollout of **HDFlow** planner on the **one_leg** assembly task initialized with **Low** randomness.

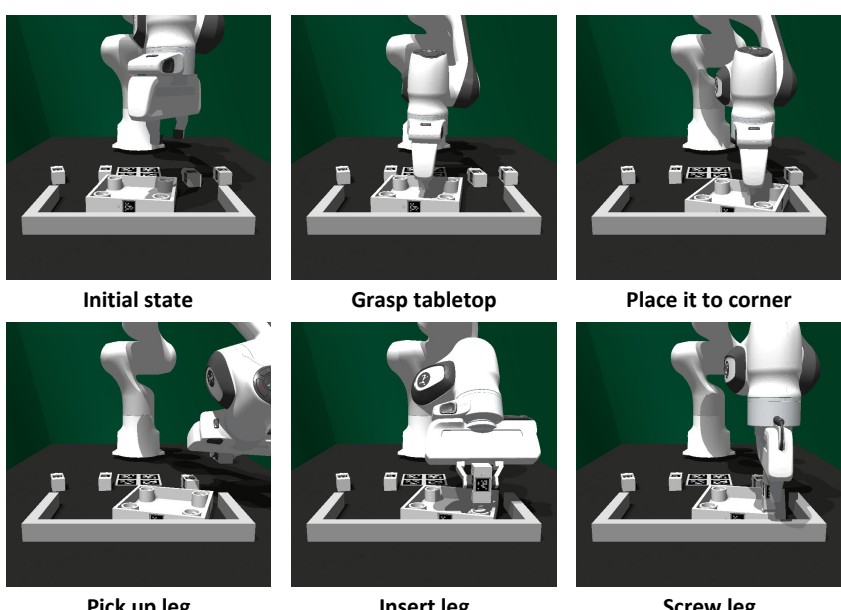

**Initial state**    **Grasp tabletop**    **Place it to corner**

**Pick up leg**    **Insert leg**    **Screw leg**

Figure 8: A successful rollout of **HDFlow** planner on the **one_leg** assembly task initialized with **Med** randomness.

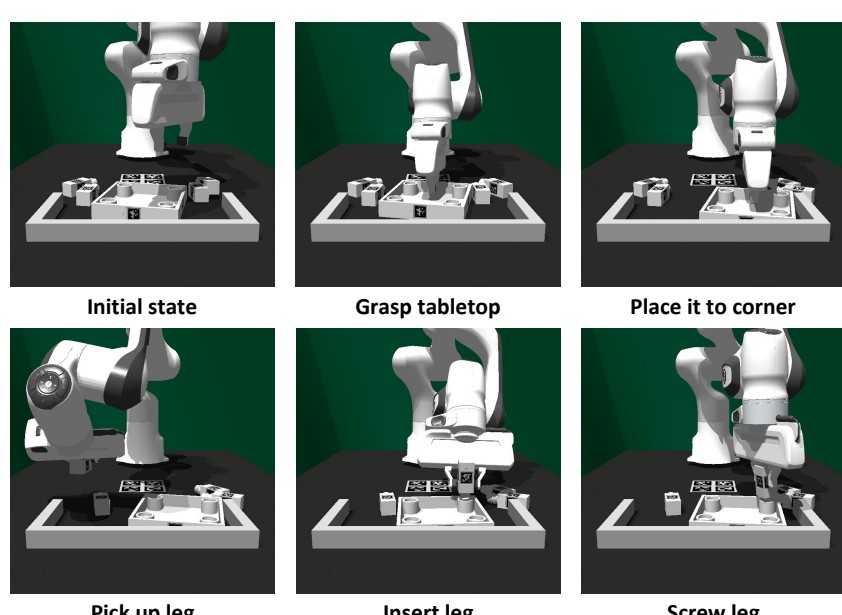

Figure 9: A successful rollout of **HDFlow** planner on the **one_leg** assembly task initialized with **High** randomness.

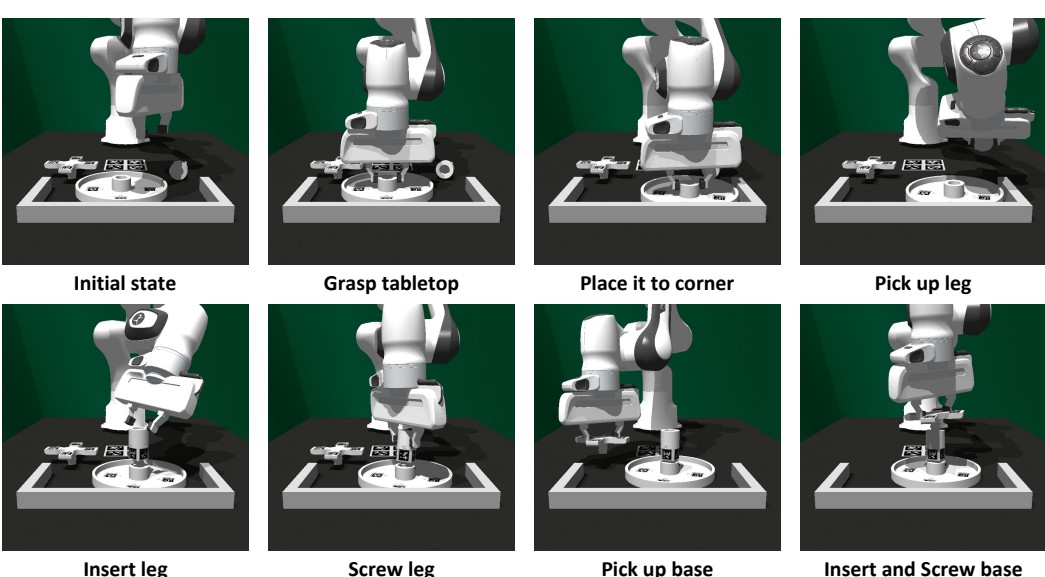

Figure 10: A successful rollout of **HDFlow** planner on the **round_table** assembly task initialized with **Low** randomness.

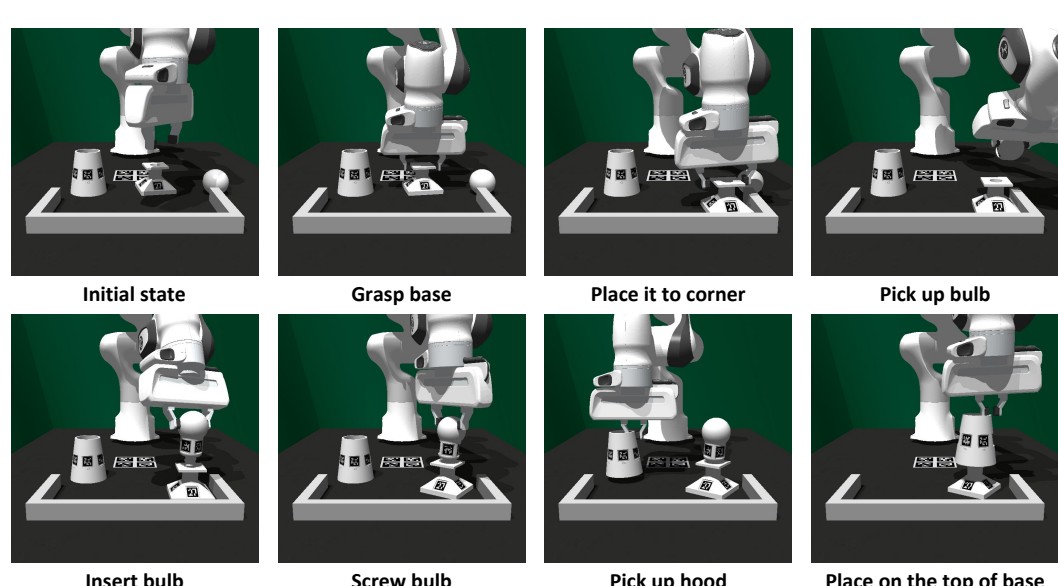

Figure 11: A successful rollout of **HDFlow** planner on the **lamp** assembly task initialized with **Low** randomness.

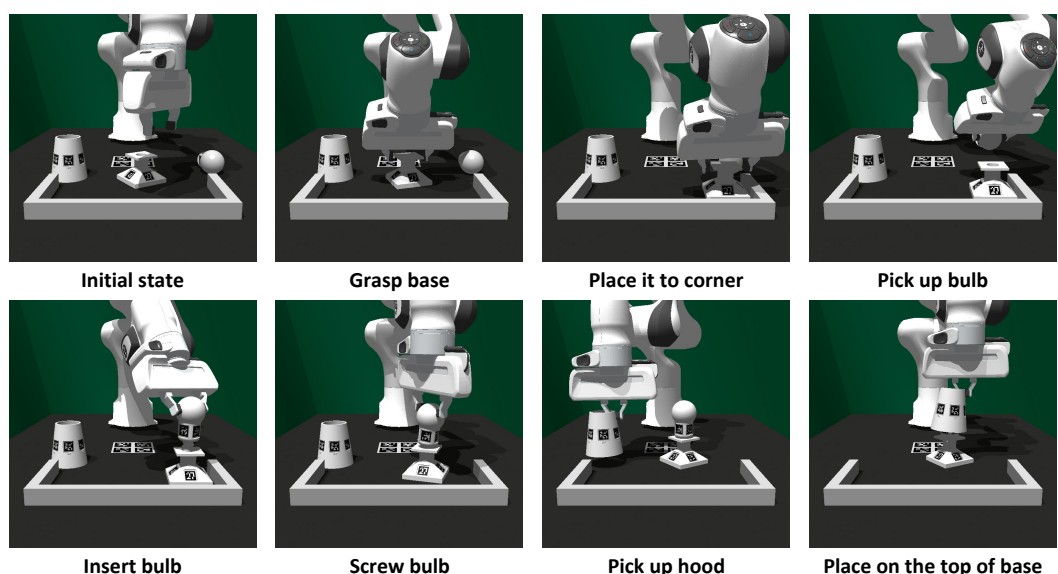

Figure 12: A successful rollout of **HDFlow** planner on the **lamp** assembly task initialized with **Med** randomness.

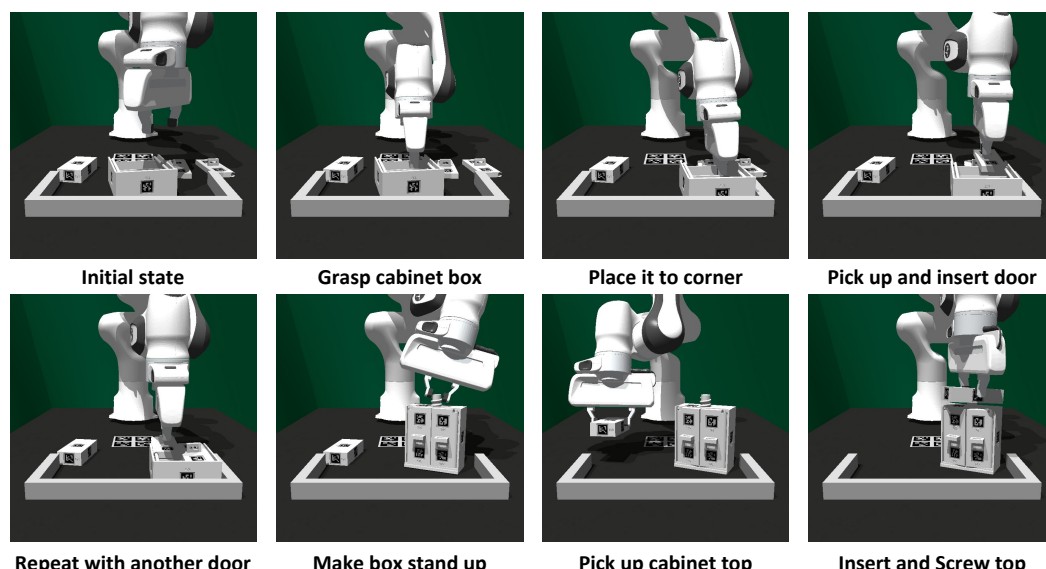

| Initial state | Grasp cabinet box | Place it to corner | Pick up and insert door |
|---|---|---|---|
| Repeat with another door | Make box stand up | Pick up cabinet top | Insert and Screw top |

Figure 13: A successful rollout of **HDFlow** planner on the **cabinet** assembly task initialized with **Low** randomness.

## I  REAL-WORLD ROLLOUTS

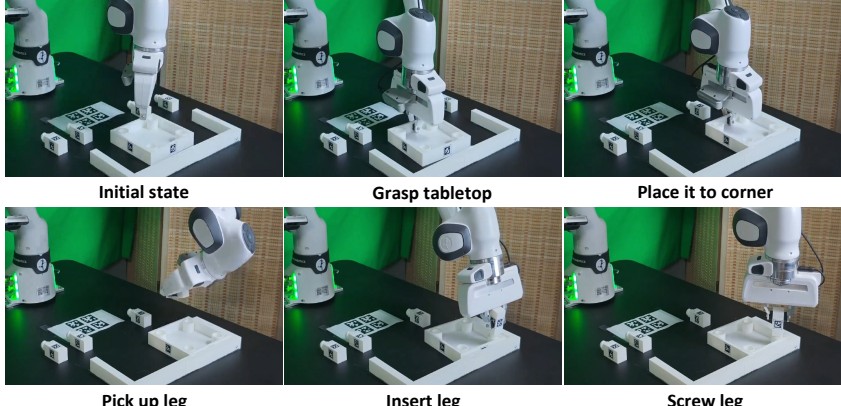

| Initial state | Grasp tabletop | Place it to corner |
|---|---|---|
| Pick up leg | Insert leg | Screw leg |

Figure 14: A successful rollout of **HDFlow** planner on the **one_leg** assembly task initialized with **Low** randomness in the real-world.

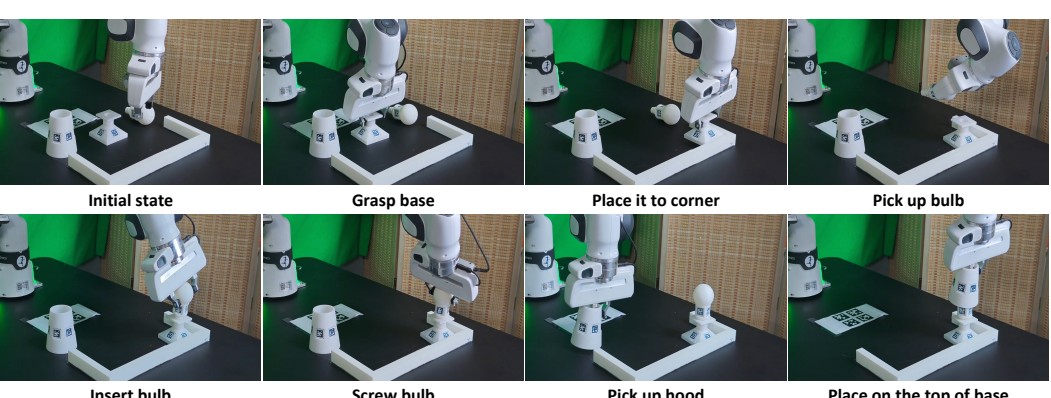

Figure 15: A successful rollout of **HDFlow** planner on the **lamp** assembly task initialized with **Low** randomness in the real-world.

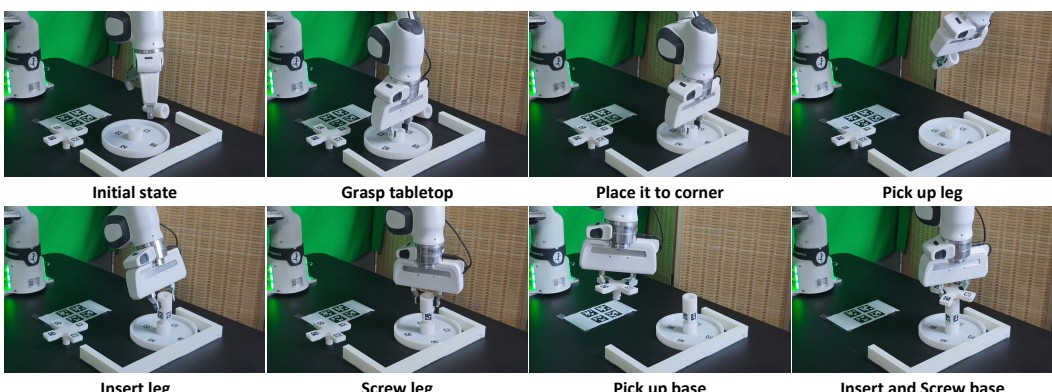

Figure 16: A successful rollout of **HDFlow** planner on the **round_table** assembly task initialized with **Low** randomness in the real-world.

