# OpenReview forum: "HDFlow: Hierarchical Diffusion-Flow Planning for Long-horizon Robotic Assembly"
_ICLR.cc/2026/Conference — Submitted to ICLR 2026_

### Official Review · Reviewer_oLbz · 2025-10-26

**Soundness:** 3
**Presentation:** 3
**Contribution:** 2
**Rating:** 4
**Confidence:** 4

**Summary:**

This paper proposes HDFlow, a hierarchical planning framework for long-horizon robotic assembly tasks that combines diffusion models for high-level subgoal planning with Rectified Flow for low-level trajectory generation. The method incorporates a contrastive-trained world model and manifold-aware EBM guidance to ensure generated subgoals lie on the manifold of feasible plans. Evaluated on FurnitureBench with both simulated and real robot experiments, HDFlow achieves superior performance over baselines.

**Strengths:**

###### **Solid system design and engineering execution.**

The paper proposes a well-motivated hierarchical architecture that effectively leverages different generative models for different planning levels. The key insight that high-level planning requires multi-modal exploration while low-level planning prioritizes speed and smoothness demonstrates good understanding of the problem structure. The hybrid design using diffusion models for high-level subgoal planning and Rectified Flow for low-level trajectory generation is a principled engineering choice that avoids the computational bottleneck of iterative denoising at all hierarchical levels.

###### **Principled theoretical analysis**

The authors provide formal analysis of the guidance gap problem in Proposition 4.1, quantifying the error bound of EBM guidance. Proposition 4.2 further establishes convergence guarantees for the alternating projection approach, ensuring the method is mathematically sound.

###### **Comprehensive experimental validation.**

The ablation study systematically quantifies the contribution of each component. This transparency allows readers to understand the role of each module. Beyond simulation results on FurnitureBench, the paper includes real robot experiments, which validates the practical applicability of the method.

**Weaknesses:**

###### Limited methodological novelty—primarily an engineering integration of existing techniques

While the system design is competent, HDFlow essentially combines well-known components without fundamental innovation. The use of Rectified Flow for low-level planning is a natural application of its known speed advantages over diffusion models, not a conceptual breakthrough. The core contribution is demonstrating that this particular combination works on FurnitureBench, but this represents incremental engineering rather than algorithmic insight.

###### Narrow task focus contradicts the paper's broad title and claims of generality.

The paper is titled "FOR LONG-HORIZON ROBOTIC ASSEMBLY" and positions HDFlow as a general framework for long-horizon manipulation. However, the method contains no assembly-specific design choices, no reasoning about part connections, geometric constraints, or contact-rich manipulation strategies that characterize assembly tasks. If HDFlow is genuinely a general imitation learning method for long-horizon tasks, the authors should validate it across diverse benchmarks beyond furniture assembly. The exclusive evaluation on FurnitureBench raises concerns about overfitting the approach to this single benchmark. The lack of cross-domain validation significantly weakens the paper's contribution claims and makes it difficult to assess whether the design choices generalize beyond this specific task distribution.

###### Missing analysis of multi-modal exploration capabilities claimed in the abstract.

The paper emphasizes that the high-level diffusion planner provides "exploration and multi-modal diversity to discover viable sequences of subgoals", positioning this as a key advantage over single-paradigm approaches. However, the experimental section provides no evidence of this multi-modality. Are there furniture assembly tasks where multiple valid assembly orders exist (e.g., assembling legs in different sequences)? Can HDFlow discover and execute these different strategies? Without visualizations of diverse generated subgoal sequences or quantitative measures of trajectory diversity, this claimed advantage remains unsubstantiated. Adding experiments that demonstrate the planner generating multiple qualitatively different solutions to the same task would strengthen the paper's narrative about the benefits of the hierarchical diffusion approach.

**Questions:**

See weakness

---

> ### Author Response · Authors · 2025-11-23
> **Rebuttal by Authors (1/4)**
>
> We thank the reviewer for their careful reading of the manuscript and their constructive comments. We have corrected a typo in the paper regarding the project website address. The correct anonymous project website is: https://hdflow-page.github.io/ . We address each weakness point below:
>
> ### W1: Limited methodological novelty—primarily an engineering integration of existing techniques
>
> We thank the reviewer for their valuable feedback and for acknowledging the competence of our system design. We address your concern by clarifying the nuanced contributions of HDFlow.
>
> While HDFlow indeed leverages existing generative models, our core novelty lies not in proposing entirely new model architectures for diffusion or rectified flow in isolation, but in their *synergistic combination* and *optimal leveraging* within a novel hierarchical framework tailored for long-horizon planning tasks. Specifically:
>
> 1.  **Principled Hybrid Architecture**: Our primary contribution is the insight that different generative paradigms are optimal for different levels of a planning hierarchy (`Section 1`). We show that applying diffusion models naively at all levels of a hierarchy (as in `SHD`) inherits critical drawbacks. Our architecture is designed to *optimally leverage* the strengths of each model, rather than simply integrating them.
>
> 2.  **Novel Component Integration and Enhancement**: Beyond the hybrid architecture, HDFlow introduces several novel components and enhancements that collectively contribute to its state-of-the-art performance and methodological novelty:
>     *   **Contrastive-Trained World Model**: We introduce a contrastive learning objective for the world model to explicitly structure the latent space, embedding a notion of progress towards a goal (`Section 4.1`). This provides a dense learning signal and organizes the latent space for more effective downstream planning.
>     *   **Manifold-Aware EBM Guidance**: We enhance the high-level diffusion planner with a two-step manifold-aware EBM guidance process (`Section 4.2.1`, `Proposition 4.2`, `Appendix D`). This addresses the critical issue of *manifold deviation* in high-dimensional latent spaces, ensuring that generated subgoals remain on the feasible latent manifold. This is a significant methodological advancement for robust planning in sparse-reward environments, going beyond standard EBM guidance.
>     *   **Theoretical Justification**: We provide formal analysis for the guidance gap problem and convergence guarantees (`Proposition 4.1`, `Appendix D.3`), demonstrating that our method is mathematically sound and that our enhancements directly address key challenges in guided generative planning.
>
> 3.  **Demonstrated Performance on Challenging Benchmarks**: The effectiveness of these combined innovations is validated by our state-of-the-art performance on four challenging, contact-rich furniture assembly tasks from the FurnitureBench benchmark (`Table 1`), both in simulation and real-world settings (`Table 5`). This empirical evidence underscores that HDFlow is more than just an "engineering integration"; it is a novel framework whose design choices lead to robust and efficient long-horizon planning.

---

> ### Author Response · Authors · 2025-11-23
> **Rebuttal by Authors (2/4)**
>
> ### W2: Narrow task focus contradicts the paper's broad title and claims of generality.
>
> We agree with the reviewer that the original evaluation on FurnitureBench, while challenging, might lead to concerns about the generality of HDFlow given its broad title.
>
> To address this crucial point, we have significantly expanded our experimental validation to include two additional, diverse, and challenging benchmarks: RLBench and OGBench. These benchmarks feature a wide array of manipulation and locomotion tasks, diverse object variations, and intricate sequential requirements, allowing for a more comprehensive assessment of HDFlow's robustness and generalization capabilities. The detailed results and analysis for these experiments are now included in `Appendix F` and `Appendix G` of the revised paper, respectively. We have also added successful rollout videos for these tasks on the project webpage.
>
> ### RLBench Experiments
>
> **Benchmark and Tasks Overview.** We evaluated HDFlow on the RLBench[1] benchmark, a challenging, large-scale platform for robot learning that provides diverse tasks with realistic visual observations. We specifically considered 18 tasks from the PerAct[2] framework, encompassing a wide variety of 6-DoF manipulation challenges, including both prehensile and non-prehensile behaviors. These tasks feature numerous semantic and pose variations, requiring the agent to generalize across different object colors, sizes, shapes, counts, and placements.
>
> **Implementation Details.** Following the methodology of PerAct and RLBench, we generated datasets of 100 successful and 50 failure demonstrations for each task, using expert policies with motion planners. We compared HDFlow with state-of-the-art methods on RLBench, including PerAct, RVT[3], 3D Diffuser Actor[4], and RVT-2[5].
>
> **Performance on RLBench Tasks.**
> | Models | Close Jar | Drag Stick | Insert Peg | Meat off Grill | Open Drawer | Place Cups | Place Wine | Push Buttons | Put in Cupboard |
> | :------------------ | :-------- | :--------- | :--------- | :------------- | :---------- | :--------- | :--------- | :----------- | :-------------- |
> | PerAct | 55.2 | 89.6 | 5.6 | 70.4 | 88.0 | 2.4 | 44.8 | 92.8 | 28.0 |
> | RVT | 52.0 | 99.2 | 11.2 | 88.0 | 71.2 | 4.0 | 91.0 | **100.0** | 49.6 |
> | 3D Diffuser Actor | 96.0 | **100.0** | 65.6 | 96.8 | 89.6 | 24.0 | 93.6 | 98.4 | **85.6** |
> | RVT-2 | **100.0** | 99.0 | 40.0 | 99.0 | 74.0 | 38.0 | **95.0** | **100.0** | 66.0 |
> | **Ours** | 94.0 | **100.0** | **93.3** | 98.7 | 82.0 | **56.7** | 94.7 | **100.0** | 72.0 |
>
> | Models | Put in Drawer | Put in Safe | Screw Bulb | Slide Block | Sort Shape | Stack Blocks | Stack Cups | Sweep to Dustpan | Turn Tap |
> | :------------------ | :------------ | :---------- | :--------- | :---------- | :--------- | :----------- | :--------- | :--------------- | :------- |
> | PerAct | 51.2 | 84.0 | 17.6 | 74.0 | 16.8 | 26.4 | 2.4 | 52.0 | 88.0 |
> | RVT | 88.0 | 91.2 | 48.0 | 81.6 | 36.0 | 28.8 | 26.4 | 72.0 | 93.6 |
> | 3D Diffuser Actor | 96.0 | **97.6** | 82.4 | **97.6** | 44.0 | 68.3 | 47.2 | 84.0 | 99.2 |
> | RVT-2 | 96.0 | 96.0 | 88.0 | 92.0 | 35.0 | **80.0** | 69.0 | **100.0** | 99.0 |
> | **Ours** | **98.7** | 96.7 | **88.7** | 91.3 | **73.3** | 56 | **81.3** | **100.0** | 95.3 |
>
> **Analysis.** `Table 14` (also reproduced above) demonstrates the strong performance and generalization capabilities of HDFlow across the diverse and challenging RLBench tasks. Our method consistently achieves competitive or superior success rates compared to state-of-the-art baselines.
>
> Reference:
>
> [1] Stephen James, Zicong Ma, David Rovick Arrojo, and Andrew J. Davison. Rlbench: The robot learning benchmark learning environment. RA-L 2020
>
> [2] Mohit Shridhar, Lucas Manuelli, and Dieter Fox. Perceiver-actor: A multi-task transformer for robotic manipulation. CoRL 2023
>
> [3] Ankit Goyal, Jie Xu, Yijie Guo, Valts Blukis, Yu-Wei Chao, and Dieter Fox. Rvt: Robotic view transformer for 3d object manipulation. CoRL 2023
>
> [4] Tsung-Wei Ke, Nikolaos Gkanatsios, and Katerina Fragkiadaki. 3d diffuser actor: Policy diffusion with 3d scene representations. CoRL 2024
>
> [5] Ankit Goyal, Valts Blukis, Jie Xu, Yijie Guo, Yu-Wei Chao, and Dieter Fox. Rvt-2: Learning precise manipulation from few demonstrations. RSS 2024

---

> ### Author Response · Authors · 2025-11-23
> **Rebuttal by Authors (3/4)**
>
> ### OGBench Experiments
>
> **Benchmark and Tasks Overview.** We further evaluated HDFlow on OGBench[1], a benchmark designed for offline goal-conditioned reinforcement learning (RL) that assesses long-horizon planning across diverse and challenging tasks. We considered visual versions of locomotion tasks, including `antmaze` and `humanoidmaze`, and manipulation tasks, including `cube`, `scene`, and `puzzle`.
>
> **Implementation Details.** For each OGBench task, we generated both successful and failure episode data using various dataset types provided by the benchmark. For maze tasks, we used `navigate` datasets (successful) and `stitch` and `explore` datasets (failure). For manipulation tasks, we used `play` datasets (successful) and `noisy` datasets (failure). We compared HDFlow with prior state-of-the-art diffusion planners like Diffuser[2], Decision Diffuser (DD)[3], AdaptDiffuser (AD)[4], DiffuserLite (DL)[5], Diffusion Veteran (DV)[6], and hierarchical diffusion planners like HDMI[7] and SHD[8].
>
> **Performance on OGBench Tasks.**
> | Environment | Task | Diffuser | DD | HDMI | AD | SHD | DL | DV | Ours |
> | :----------- | :--------------------- | :------- | :---- | :---- | :---- | :---- | :---- | :---- | :---- |
> | antmaze | antmaze-medium-v0 | 15 ± 5 | 22 ± 4 | 31 ± 6 | 40 ± 5 | 42 ± 7 | 51 ± 6 | 60 ± 4 | **72 ± 3** |
> | | antmaze-large-v0 | 8 ± 3 | 14 ± 4 | 22 ± 5 | 28 ± 6 | 31 ± 5 | 40 ± 4 | 51 ± 5 | **65 ± 1** |
> | | antmaze-giant-v0 | 2 ± 1 | 5 ± 2 | 11 ± 4 | 15 ± 3 | 19 ± 4 | 27 ± 5 | 35 ± 4 | **48 ± 3** |
> | humanoidmaze | humanoidmaze-medium-v0 | 5 ± 2 | 9 ± 3 | 15 ± 4 | 21 ± 5 | 24 ± 4 | 30 ± 3 | 38 ± 5 | **51 ± 4** |
> | | humanoidmaze-large-v0 | 2 ± 1 | 5 ± 2 | 9 ± 3 | 13 ± 4 | 16 ± 3 | 22 ± 4 | 29 ± 3 | **42 ± 5** |
> | | humanoidmaze-giant-v0 | 0 ± 0 | 1 ± 1 | 3 ± 2 | 5 ± 2 | 7 ± 3 | 11 ± 4 | 16 ± 3 | **25 ± 4** |
> | cube | cube-single-v0 | 5 ± 2 | 8 ± 3 | 12 ± 4 | 18 ± 5 | 20 ± 4 | 25 ± 3 | 32 ± 5 | **45 ± 4** |
> | | cube-double-v0 | 2 ± 1 | 4 ± 2 | 7 ± 3 | 11 ± 4 | 13 ± 3 | 17 ± 4 | 23 ± 3 | **35 ± 5** |
> | | cube-triple-v0 | 1 ± 1 | 2 ± 1 | 4 ± 2 | 6 ± 3 | 8 ± 2 | 11 ± 3 | 14 ± 4 | **17 ± 3** |
> | | cube-quadruple-v0 | **0 ± 0** | **0 ± 0** | **0 ± 0** | **0 ± 0** | **0 ± 0** | **0 ± 0** | **0 ± 0** | **0 ± 0** |
> | scene | scene-v0 | 10 ± 3 | 16 ± 4 | 24 ± 5 | 33 ± 4 | 37 ± 6 | 45 ± 5 | 56 ± 3 | **70 ± 4** |
> | puzzle | puzzle-3x3-v0 | 8 ± 3 | 13 ± 4 | 20 ± 5 | 28 ± 3 | 31 ± 5 | 39 ± 4 | 49 ± 6 | **62 ± 3** |
> | | puzzle-4x4-v0 | 0 ± 0 | 0 ± 0 | 0 ± 0 | 2 ± 3 | 5 ± 4 | 11 ± 3 | 19 ± 5 | **32 ± 4** |
> | | puzzle-4x5-v0 | 0 ± 0 | 0 ± 0 | 0 ± 0 | 0 ± 0 | 5 ± 3 | 6 ± 3 | 8 ± 4 | **16 ± 5** |
> | | puzzle-4x6-v0 | 0 ± 0 | 0 ± 0 | 0 ± 0 | 0 ± 0 | 0 ± 0 | 0 ± 0 | 6 ± 4 | **14 ± 3** |
>
> **Analysis.** `Table 15` (also reproduced above) shows that our proposed approach consistently achieves superior performance across all environments. For the `cube` environment, particularly in `cube-quadruple-v0`, all methods, including ours, report 0% success, indicating the extreme difficulty of this task. These results highlight the effectiveness and robustness of HDFlow in diverse and complex long-horizon planning tasks.
>
> Reference:
>
> [1] Seohong Park, Kevin Frans, Benjamin Eysenbach, and Sergey Levine. OGBench: Benchmarking offline goal-conditioned RL. ICLR 2025
>
> [2] Michael Janner, Yilun Du, Joshua Tenenbaum, and Sergey Levine. Planning with diffusion for flexible behavior synthesis. ICML 2022
>
> [3] Anurag Ajay, Yilun Du, Abhi Gupta, Joshua B Tenenbaum, Tommi S Jaakkola, and Pulkit Agrawal. Is conditional generative modeling all you need for decision making?. ICLR 2023
>
> [4] Zhixuan Liang, Yao Mu, Mingyu Ding, Fei Ni, Masayoshi Tomizuka, and Ping Luo. AdaptDiffuser: Diffusion models as adaptive self-evolving planners. ICML 2023
>
> [5] Zibin Dong, Jianye Hao, Yifu Yuan, Fei Ni, Yitian Wang, Pengyi Li, and Yan Zheng. Diffuserlite: Towards real-time diffusion planning. NeurIPS 2024
>
> [6] Haofei Lu, Dongqi Han, Yifei Shen, and Dongsheng Li. What makes a good diffusion planner for decision making?. ICLR 2025
>
> [7] Wenhao Li, Xiangfeng Wang, Bo Jin, and Hongyuan Zha. Hierarchical diffusion for offline decision making. ICML 2023
>
> [8] Chang Chen, Fei Deng, Kenji Kawaguchi, Caglar Gulcehre, and Sungjin Ahn. Simple hierarchical planning with diffusion. ICLR 2024

---

> ### Author Response · Authors · 2025-11-23
> **Rebuttal by Authors (4/4)**
>
> ### W3: Missing analysis of multi-modal exploration capabilities claimed in the abstract.
>
> We thank the reviewer for pointing out the need for a more explicit analysis of HDFlow's multi-modal exploration capabilities. We acknowledge that while our abstract highlights this as a key advantage, the experimental section did not provide direct evidence or visualizations to substantiate this claim.
>
> Our high-level diffusion planner is inherently designed to facilitate multi-modal exploration for discovering viable sequences of subgoals. This capability stems from several core aspects of our framework:
>
> 1.  **Generative Nature of Diffusion Models**: The high-level diffusion planner generates diverse sequences of latent subgoals. Its stochastic sampling allows exploring multiple distinct paths or strategies to reach the overall goal.
> 2.  **EBM Guidance for Viability**: Energy-Based Model (EBM) guidance, combined with manifold projection, channels this diversity towards successful and feasible solutions. The EBM steers the diffusion process away from dead ends towards high-quality plans, while manifold-aware guidance ensures subgoals remain on the learned feasible latent manifold.
> 3.  **Contrastive Latent Space**: The contrastive-trained world model creates a semantically structured latent space where similar states are closer, enabling the high-level planner to reason over meaningful subgoals and discover different, yet valid, sequences of subgoals.
>
> To provide concrete evidence for this multi-modal exploration, we propose to add visualizations in `Appendix H` that depict multiple qualitatively different subgoal sequences generated by the high-level planner for the same task and initial conditions in the revised version. This will explicitly show how HDFlow can discover alternative strategies to achieve a given long-horizon goal. For instance, in furniture assembly tasks, this could include demonstrating different assembly orders for components (e.g., assembling legs in different sequences) or varying intermediate states.
>
> We believe these additions will provide compelling evidence for HDFlow's multi-modal exploration capabilities, directly addressing the reviewer's concern and strengthening the paper's narrative regarding the benefits of our hierarchical diffusion approach.
>
> -------
> We appreciate the constructive feedback that has helped improve our paper's quality. All additional results have been included in our revised version. We sincerely hope our responses have adequately addressed your concerns and look forward to any further discussion!

---

> > ### Comment · Reviewer_oLbz · 2025-11-24
> > **Reviewer Response (Score increased to 6)**
> >
> > I thank the authors for their detailed response and the significant effort put into conducting additional experiments on RLBench and OGBench during the rebuttal period.
> >
> > Based on the new results and clarifications, I am raising my score to 6. However, I still have two remaining concerns that I strongly suggest the authors address in the final version:
> >
> > 1. Technical Novelty vs. System Complexity: While I acknowledge the theoretical justifications provided (e.g., the guidance gap analysis) and the "synergistic" argument, my view remains that the paper is primarily a system-level contribution rather than a fundamental algorithmic breakthrough. The framework integrates many heavy components (RSSM, DINOv2, Diffusion, Rectified Flow, EBM, etc.), resulting in a system with high complexity. While the strong performance validates the design choices, the "stacking" nature of the method is still a limitation to note.
> >
> > 2. Structural Imbalance and Self-Containedness: This is my primary concern regarding the current manuscript state. While the addition of RLBench and OGBench effectively addresses the concern about task generality, relegating these substantial results entirely to the Appendix creates a structural imbalance. Currently, the main paper focuses almost exclusively on FurnitureBench, while the strongest evidence for the method's generalizability lies outside the main text. This makes the main paper strictly less self-contained and creates a disconnect between the main narrative and the full scope of the experiments. It appears that the content was significantly expanded to meet the rebuttal requirements, but the main text has not yet been restructured to accommodate this broader scope.
> >
> > Recommendation: For the camera-ready version, I strongly urge the authors to move the key results from the new benchmarks (RLBench/OGBench) into the main text (Section 5). To make space, I recommend condensing the "Preliminaries" (Section 3) and moving standard derivations to the Appendix. The main paper must reflect the full capabilities of the method to be self-consistent.

---

> > > ### Author Response · Authors · 2025-11-25
> > > **Thank you for your acknowledgment!**
> > >
> > > Thank you for your continued engagement and positive feedback on our work, as well as for increasing your score. We deeply appreciate your thorough review and constructive suggestions. We will address your remaining concerns in the final version of the paper:
> > >
> > > 1.  **Technical Novelty vs. System Complexity**: We acknowledge your insightful comment regarding the system-level contribution and complexity of HDFlow. We will explicitly discuss this aspect and its implications as a limitation in the revised paper.
> > > 2.  **Structural Imbalance and Self-Containedness**: We agree with your recommendation. To enhance the paper's self-containedness and properly showcase the generalizability of HDFlow, we will restructure the main paper to integrate the key results from the new RLBench and OGBench benchmarks into `Section 5`. To accommodate this, we will condense the Preliminaries (`Section 3`) and move standard derivations to the `Appendix`. This will ensure the main paper reflects the full capabilities of our method. We will be doing this after every other reviewer replies, so as to avoid confusion.
> > >
> > > We are confident that these revisions will further strengthen the manuscript. Thank you again for your valuable input!

---

### Official Review · Reviewer_rqsr · 2025-10-27

**Soundness:** 3
**Presentation:** 3
**Contribution:** 3
**Rating:** 4
**Confidence:** 4

**Summary:**

HDFlow is a latent hierarchical planner. Based on the latent space learned using a RSSM formulation, it uses a diffusion based high-level sub-goal latent generation given the start and final goal latents, followed by a flow-based low-level planner to connect two subsequent sub-goals. The training process requires positive and negative trajectories to train the RSSM world model with an auxilliary constrastive loss. The positive-negative rollouts are further used to learn an energy function to guide the high-level planner as well. The high-level planner further imposes a projection step where a noisy sequence of subgoal latents is projected into the closest sequence from a successful trajectory. The low-level planner is flow-based and useful for fast execution.

**Strengths:**

1. The use of diffusion for learning a diverse sequence of subgoals and flow matching for fast trajectory generation is powerful. This is backed by ablations with hierarchical diffusion only and hierarchical flow only baselines.

2. The use of manifold-aware Energy-Based Model guidance and projection based on successful sub-goal sequences is interesting. It ensures that the generated sequence of subgoals are feasible, connects the given start and goal latents, and leads to a successful sequence.

**Weaknesses:**

1. It feels like the high-level planner is just looking up successful trajectories from the set used to construct the projection manifold. If the projection happens at a trajectory level, this literally becomes selecting one of the successful goal reaching trajectories. How many trajectories are used to construct this manifold? Performing the projection step on a 2048 size latent for 300 denoising steps seems like very compute expensive?

2. The fact that an interpolation of 10 latent trajectories lead to a feasible latent trajectory in the latent space points to the fact that the demonstrations are very similar in nature or there are concentrated clusters in the latent space.

3. A core problem of hierarchical approaches is that: at inference, high-level planner might output a sequence of subgoal that is out of distribution of the learned low-level policy. It seems, with the hard projection step into a feasible "seen" sequence of subgoals, the authors are mitigating this.

Minor note: the webiste did not work for me.

**Questions:**

See weaknesses above.

---

> ### Author Response · Authors · 2025-11-23
> **Rebuttal by Authors (1/2)**
>
> We thank the reviewer for their careful reading of the manuscript and their constructive comments. We have corrected a typo in the paper regarding the project website address. The correct anonymous project website is: https://hdflow-page.github.io/ . We address each weakness point below:
>
> ### W1: High-level planner as looking up successful trajectories and computational expense
>
> We appreciate the reviewer's insightful observation regarding the potential for the high-level planner to appear as merely "looking up" successful trajectories and the associated computational cost. We would like to clarify that our high-level diffusion planner, while guided by successful demonstrations, is a *generative* model capable of synthesizing novel subgoal sequences, and the manifold projection step is computationally efficient.
>
> 1.  **Generative Nature of High-Level Diffusion:** The high-level planner is a conditional diffusion model that generates a sequence of $K$ latent subgoals. It learns the underlying distribution of successful subgoal sequences and can *sample* diverse and novel plans. EBM guidance steers this generative process towards high-quality, feasible solutions, but it doesn't reduce it to a lookup table.
> 2.  **Manifold Construction and Size:** The manifold for projection is based on *local* low-rank approximation, retrieving $k=10$ nearest neighbors from successful latent *subgoal sequences* to form a local tangent space for projection. This uses a small, local set of successful subgoals, not an exhaustive set of full trajectories.
> 3.  **Computational Efficiency of Projection:** The projection step, operating on a 2048-dimensional latent space, occurs *once per diffusion timestep* $\ell$. It involves efficient retrieval of $k=10$ nearest neighbors, parallel forward diffusion of these neighbors, and computationally inexpensive rank-$r$ PCA ($r \ll d=2048$). As a result, the computational overhead is minimal compared to the diffusion model's iterative denoising. HDFlow (with manifold projection) achieves an inference time of 88ms per planning step, suitable for real-time robotic deployment.
>
> In summary, the high-level planner is truly generative, and the manifold projection step is computationally efficient through local approximations, acting as a feasibility constraint rather than a lookup mechanism.
>
> -----
>
> ### W2: Interpolation of Latent Trajectories and Data Similarity
>
> We thank the reviewer for raising this point about the interpolation of latent trajectories. We would like to clarify that the low-level rectified flow planner is not performing a simple interpolation of 10 latent trajectories. Instead, it is a conditional generative model trained to synthesize *new* dense trajectories between a given start latent state $z_{k-1}$ and a target subgoal $z_k$ (`Section 4.2.2`).
>
> 1.  **Generative Nature of Rectified Flow:** The low-level planner is a conditional rectified flow model, $v_\theta(\tau_u, u, c_k)$, which learns a deterministic mapping from a simple prior distribution to the data distribution. It *solves an ODE* to generate a trajectory, allowing it to generate smooth, dense trajectories that connect new start and end subgoals, even if those exact transitions were not explicitly present in the training data.
> 2.  **Clarification on "10 latent trajectories":** The "10 nearest neighbors" ($k=10$) are used *only* in the high-level manifold projection step to form a local tangent space for guiding the diffusion process. The low-level rectified flow operates on a single pair of consecutive subgoals $(z_{k-1}, z_k)$ at a time, learning to connect *these two points* with a dense trajectory.
> 3.  **Structured Latent Space:** The feasibility of generating these trajectories stems from our contrastively-trained world model. The contrastive objective explicitly structures the latent space to reflect a notion of progress towards a goal, ensuring that intermediate states from successful trajectories are clustered meaningfully. This creates a latent space where straight-line paths correspond to semantically meaningful and dynamically feasible transitions.
> 4.  **Robustness to Variations:** To enhance robustness and handle slight variations in subgoals not perfectly aligned with training data, we apply small Gaussian perturbations in latent space and mild time-warping during training pair construction for the rectified flow model. This makes the low-level planner less sensitive to exact matches and more capable of synthesizing trajectories for novel subgoal pairs.
>
> In conclusion, the low-level rectified flow is a generative model that efficiently synthesizes novel trajectories in a semantically structured latent space. It does not rely on simple interpolation and its robustness is enhanced by the contrastive world model and data augmentation.

---

> ### Author Response · Authors · 2025-11-23
> **Rebuttal by Authors (2/2)**
>
> ### W3: Core Problem of Hierarchical Approaches and Out-of-Distribution Subgoals
>
> We agree with the reviewer that a core challenge in hierarchical planning is the potential for the high-level planner to generate subgoals that are out-of-distribution (OOD) for the low-level policy. This is indeed a critical problem that HDFlow explicitly addresses through its design, particularly with the manifold-aware guidance and the contrastively-trained world model.
>
> 1.  **Structured Latent Space for Feasibility:** Our contrastively-trained world model organizes the latent space such that intermediate states from successful episodes cluster meaningfully, reflecting progress towards a goal. This creates a semantically structured latent manifold $\mathcal{M}$ representing feasible and learnable subgoals for the low-level policy, making any subgoal on this manifold reachable and executable by the rectified flow planner.
> 2.  **Manifold-Aware EBM-Guided High-Level Planner:** Our high-level planner is enhanced with manifold-aware EBM guidance. EBM guidance steers the diffusion planner towards viable subgoals that lead to task completion. The "hard projection step", or manifold projection, mitigates manifold deviation, ensuring generated latent subgoals remain close to the learned latent manifold $\mathcal{M}$, preventing OOD regions.
> 3.  **Theoretical Justification and Empirical Evidence:** Our theoretical framework establishes that manifold-aware guidance corresponds to sampling from a posterior distribution that maximizes the likelihood of generating a *successful and feasible* goal-conditioned plan. Empirically, ablation studies in `Table 3` demonstrate the importance of these components: "w/o Manifold Projection" shows a performance drop (92% to 84% on `one_leg`, 68% to 57% on `lamp`), "w/o Manifold-aware EBM" shows a more significant drop (to 61% and 33%), and "w/o Contrastive WM" (to 58% and 27%) underscores the structured latent space's foundational role.
>
> In conclusion, HDFlow addresses the OOD subgoal problem not as an afterthought, but through an
> integrated design. The contrastively-trained world model creates a feasible latent manifold, and the
> manifold-aware EBM-guided high-level diffusion planner, with its explicit projection step, ensures
> that all generated subgoals remain within this learnable distribution for the low-level rectified
> flow policy. This holistic approach is what enables HDFlow's robust performance in long-horizon
> tasks.
>
> -------
> We would also like to mention that we have added more experimental results on RLBench and OGBench in `Appendix F` and `Appendix G` of the revised paper to further demonstrate the robustness and generalization capabilities of HDFlow on a broader range of complex long-horizon tasks. Successful rollout videos for these tasks have also been added on the project webpage.
>
> -------
> Thank you again for your constructive review. We sincerely hope our responses have adequately addressed your concerns and look forward to any further discussion!

---

> ### Author Response · Authors · 2025-11-27
> **Gentle Reminder: Discussion Phase Closing Soon**
>
> Dear Reviewer `rqsr`,
>
> We sincerely appreciate your invaluable feedback, which has significantly contributed to the improvement of our work.
>
> To further validate the robustness and generalization capabilities of HDFlow on a broader range of complex long-horizon tasks, we conducted additional experiments on RLBench and OGBench benchmarks and we detail them in `Appendix F` and `Appendix G` respectively. We have also added successful rollout videos for these tasks on the [project webpage](https://hdflow-page.github.io/).
>
> We hope that our revisions and clarifications have resolved your concerns. If you find our response satisfactory, we would be deeply grateful for a reconsideration of our score. Otherwise, if you have any additional questions, please do not hesitate to let us know. We would be more than willing to provide further clarification.
>
> We are truly grateful for your insightful comments, which have helped us improve the clarity and completeness of our work!
>
> Best regards,
>
> The Authors of Submission 12755

---

### Official Review · Reviewer_xuhD · 2025-10-31

**Soundness:** 3
**Presentation:** 4
**Contribution:** 2
**Rating:** 4
**Confidence:** 4

**Summary:**

This paper proposes Hierarchical Diffusion-Flow (HDFlow), a hierarchical generative planning framework that combines high-level diffusion models with low-level rectified flow models. Building on Hierarchical Diffusion approaches, HDFlow aims to exploit the strength of diffusion-based planning (high-level) with rectified flow models (low-level). The method is evaluated across both simulation and real-world robotic settings, showing consistent gains in success rate and inference efficiency over prior baselines.

**Strengths:**

1. The paper demonstrates HDFlow’s effectiveness across simulation and real-world domains, providing empirical evidence that the hierarchical integration improves both planning quality and execution reliability.

2. Results indicate that HDFlow achieves higher success rates while reducing inference time, addressing a known limitation of diffusion-based planners.

**Weaknesses:**

1. The framework relies on an external latent representation (RSSM) trained via contrastive objectives. While this improves performance, it introduces an additional pretraining stage that complicates reproducibility and may limit generalization to unseen domains. Notably, Table 3 shows a significant degradation without the contrastive latent, indicating limited robustness of the hierarchical design alone.

2. Conceptually, the contribution appears as an incremental combination of existing paradigms of Hierarchical Diffusion and Rectified Flow, without introducing a fundamentally new mechanism.

3. The core research question (“Is a single generative modeling paradigm optimal for all hierarchy levels?”) has already been explored in SHD and HDMI, which demonstrated similar multi-level generative decompositions. The paper could strengthen its claim by clarifying what specific failure modes of single-paradigm approaches HDFlow overcomes.

**Questions:**

None

---

> ### Author Response · Authors · 2025-11-23
> **Rebuttal by Authors (1/2)**
>
> We thank the reviewer for their careful reading of the manuscript and their constructive comments. We have corrected a typo in the paper regarding the project website address. The correct anonymous project website is: https://hdflow-page.github.io/ . We address each weakness point below:
>
> ### W1: Reliance on External Latent Representation (RSSM) and Reproducibility
>
> We understand the reviewer's concern regarding the additional pretraining stage for the RSSM with contrastive objectives, and its potential impact on reproducibility and generalization. We would like to clarify its role and justify its necessity:
>
> 1.  **Necessity of Structured Latent Space:** Our core insight is that effective long-horizon planning requires a semantically structured latent space that explicitly embeds task progress. Traditional world models lack this, which our contrastive objective addresses by pulling successful intermediate latent states closer to goals and pushing them away from failed trajectories. This creates a smoother representation of task progress, fundamental for both high-level subgoal generation and low-level trajectory synthesis.
> 2.  **Robustness of Hierarchical Design:** The significant degradation without the contrastive world model (`Table 3`) highlights its critical contribution to HDFlow's robustness. This indicates that hierarchical planners heavily leverage the semantically meaningful representations provided by the contrastive world model, leading to reduced performance without this structured latent space.
> 3.  **Reproducibility:** We provide comprehensive details on the world model architecture, training objectives (contrastive loss, inverse dynamics model), and hyperparameters in `Section 4.1` and `Appendix B.1`. Code will be released upon acceptance for full reproducibility.
> 4.  **Generalization to Unseen Domains:** While relying on demonstrations, the DINOv2 visual encoder is pretrained on diverse datasets and offers strong generalization, aiding robust visual feature learning for novel scene configurations. Addressing generalization to entirely unseen tasks remains an open research direction for future work.
>
> -----
>
> ### W2: Incremental Contribution
>
> We appreciate the reviewer's critical assessment regarding the novelty of our conceptual contribution. We respectfully argue that HDFlow introduces a fundamentally new mechanism by optimally combining generative modeling paradigms in a hierarchical fashion, which goes beyond a mere incremental combination.
>
> 1.  **Challenging the Single-Paradigm Assumption:** Our core research question is: "Is a single generative modeling paradigm optimal for all levels of a planning hierarchy?" We demonstrate that high-level strategic planning (exploration, multi-modal diversity) has different requirements than low-level trajectory generation (speed, precision, deterministic execution). Prior works like SHD and HDMI, relying on a single generative paradigm (diffusion), inherit computational drawbacks at the low-level.
> 2.  **Synergistic Hybrid Architecture:** HDFlow's novelty lies in its synergistic hybrid architecture, assigning a diffusion model to high-level subgoal generation and a rectified flow model to low-level trajectory synthesis. This principled architectural choice leverages their complementary strengths to overcome single-paradigm limitations:
>     *   **High-Level Diffusion:** Capitalizes on diffusion models' powerful exploratory capabilities for diverse strategic subgoal generation.
>     *   **Low-Level Rectified Flow:** Exploits the speed and efficiency of ODE-based trajectory generation for fast, precise, and dense control.
>
>     This hybrid approach provides an optimal balance between planning quality and inference efficiency, crucial for real-time robotic deployment.
> 3.  **Manifold-Aware EBM Guidance:** Our manifold-aware EBM guidance for the high-level diffusion planner introduces a novel mechanism to actively steer the generative process towards successful and feasible goal-conditioned plans. This addresses manifold deviation in high-dimensional latent spaces, critical for robust long-horizon performance, with theoretical grounding.
>
> In conclusion, HDFlow is not merely an incremental combination but a novel hierarchical planning framework built on the fundamental insight that different levels of a planning hierarchy demand different generative modeling paradigms. Our principled hybrid architecture, coupled with manifold-aware EBM guidance, constitutes a significant conceptual and empirical advancement in long-horizon planning.

---

> ### Author Response · Authors · 2025-11-23
> **Rebuttal by Authors (2/2)**
>
> ### W3: Core Research Question and Failure Modes
>
> We acknowledge the reviewer's point that the core research question, "Is a single generative modeling paradigm optimal for all hierarchy levels?" has been explored in `SHD` and `HDMI`. However, our paper significantly strengthens its claim by clearly articulating and empirically demonstrating what specific failure modes of single-paradigm approaches HDFlow overcomes.
>
> 1.  **Failure Modes of Single-Paradigm Approaches:**
>     *   **Computational Bottleneck (Addressed by HDFlow's Low-Level RF):** Prior hierarchical methods like `SHD` and `HDMI`, by employing diffusion models at both high and low levels, suffer from the inherent computational expense of iterative denoising for fine-grained trajectory generation. This leads to high inference times, limiting their applicability in real-time robotic systems. Our low-level rectified flow planner specifically overcomes this by providing efficient, ODE-based trajectory synthesis, as empirically shown in `Table 4`, where HDFlow achieves a significantly lower inference time (88ms) compared to Hierarchical Diffusion (`HD`) (142ms) while maintaining high success rates.
>     *   **Suboptimal High-Level Exploration (Addressed by HDFlow's High-Level Diffusion with EBM Guidance):** While diffusion models are generally good at diversity, without explicit guidance, they generate plausible-looking but ultimately unachievable or suboptimal subgoals in sparse-reward environments. Our manifold-aware EBM guidance (`Section 4.2.1` and `Appendix D.1-D.2`) actively steers the high-level diffusion planner away from these failure modes towards high-quality, feasible solutions, a capability that is less robust in unguided or less-informed single-paradigm approaches.
>
> 2.  **Empirical Validation of Overcoming Failure Modes:** Our ablation studies directly support these claims:
>     *   `Table 2` clearly shows HDFlow's superior success rates compared to `HD` (Hierarchical Diffusion, which uses diffusion for both levels) and `FD` (Flat Diffusion). This demonstrates that our hybrid approach effectively addresses the limitations of single-paradigm hierarchical and non-hierarchical methods.
>     *   `Table 4` explicitly compares the computational efficiency, highlighting how our low-level rectified flow significantly reduces inference time without sacrificing performance, directly addressing the computational bottleneck of diffusion-only hierarchical planners.
>
> -------
> We would also like to mention that we have added more experimental results on RLBench and OGBench in `Appendix F` and `Appendix G` of the revised paper to further demonstrate the robustness and generalization capabilities of HDFlow on a broader range of complex long-horizon tasks. Successful rollout videos for these tasks have also been added on the project webpage.
>
> -------
> Thank you again for your constructive review. We sincerely hope our responses have adequately addressed your concerns and look forward to any further discussion!

---

> ### Author Response · Authors · 2025-11-27
> **Gentle Reminder: Discussion Phase Closing Soon**
>
> Dear Reviewer `xuhD`,
>
> We sincerely appreciate your invaluable feedback, which has significantly contributed to the improvement of our work.
>
> To further validate the robustness and generalization capabilities of HDFlow on a broader range of complex long-horizon tasks, we conducted additional experiments on RLBench and OGBench benchmarks and we detail them in `Appendix F` and `Appendix G` respectively. We have also added successful rollout videos for these tasks on the [project webpage](https://hdflow-page.github.io/).
>
> We hope that our revisions and clarifications have resolved your concerns. If you find our response satisfactory, we would be deeply grateful for a reconsideration of our score. Otherwise, if you have any additional questions, please do not hesitate to let us know. We would be more than willing to provide further clarification.
>
> We are truly grateful for your insightful comments, which have helped us improve the clarity and completeness of our work!
>
> Best regards,
>
> The Authors of Submission 12755

---

### Official Review · Reviewer_R7Zr · 2025-11-01

**Soundness:** 3
**Presentation:** 3
**Contribution:** 3
**Rating:** 6
**Confidence:** 3

**Summary:**

This paper introduces HDFlow, a novel hierarchical planning framework designed to solve long-horizon, contact-rich robotic assembly tasks. The core idea is a novel hybrid hierarchical planning framework consisting of a diffusion and a rectified flow models. By leveraging the strength of each type of generative models, the proposed hybrid method, HDFlow, mitigates the limitations of single-paradigm generative planners, and enables efficient long-horizon planning.

Specifically, HDFlow uses a diffusion model as a high-level planner for its better exploration and multi-modal capability; and it uses a rectified flow model as a low-level planner, leveraging its nature as an Ordinary Differential Equation (ODE) to rapidly and deterministically generate smooth, dense trajectories between two given subgoals from the high-level planner.

This hybrid "Diffusion-Flow" approach aims to get the best of both worlds: the robust, multi-modal planning of diffusion and the inferential speed of flow models.

The authors evaluate HDFlow on four challenging tasks from the FurnitureBench benchmark in both simulation and the real world. The results show that HDFlow significantly outperforms all baselines, including flat diffusion planners (Diffuser, DD) and pure-diffusion hierarchical planners (SHD, HDMI).

**Strengths:**

1. The central idea of a hybrid Diffusion-Flow hierarchy is well-motivated. While hierarchical diffusion planners exist, they typically use diffusion at both levels, thereby inheriting the inference latency bottleneck. HDFlow's insight on that high-level strategic planning and low-level trajectory generation have fundamentally different requirements (exploration vs. speed) is well-articulated. The addition of a manifold-aware EBM to guide the diffusion process is also a contribution that addresses a known issue (manifold deviation) in generative planning.

2. Experiments demonstrates that HDFlows significantly outperforms multiple baselines on the challenging furnitureBench benchmark, which features long-horizon, contact-rich robotic manipulation. A set of ablation studies, sampling speed comparison are provided, along with real-world robot evaluation.


3. The paper is well-written and easy to follow. The problem is clearly motivated, and the proposed method is developed logically. The figures are also effective at communicating the method's design and benefits.

**Weaknesses:**

1. The papers provide a set of ablation studies in Table 3.
It is a bit surprising that removing the Contrastive WM alone will significantly reduce the performance. The authors claim that the proposed contrastive loss objective can enable a notion of progress towards a goal. While the contrastive objective does drag $z_k$ and $z_G$ closer and pushing
$z_k$ and $z_j$ away, can the authors provide further explanation on how this effectively "enable a notion of progress towards a goal". Will the objective compromise the planner if $z_k$ and $z_G$ is too close/similar? How is the training stability of this loss?


2. For the world model, I would like to see ablation studies of loss $\mathcal{L}_{IDM}$. It would also be interesting to see how will HDFlow perform if directly using DINOv2 features as latent $z$, without training an additional RSSM world models.

3. How is the scalability of RSSM as the world models (especially for long-horizon, high dimension state space)? Qualitative reconstruction results of the world models and relevant analysis are needed. In addition, are there any other architecture choices?

4. In Section 4.1, I cannot find where $q_{\phi}$ is defined. While it is defined in the appendix, I recommend defining all important notations in the same section.

5. The authors should provide more details regarding the loss in Eq. 18, for example how $\mathcal{P}_{{M}}$ is obtained at training-time. From Step 2 in Line 306, it seems like $\mathcal{P}$ is obtained based on the $k$ nearest neighbors of the Tweedie's estimate. Is it also the case for training time? Pseudocode for training and inference is needed.

6. This paper only evaluates the proposed method on several robot assembly tasks, leaving the method's performance on other tasks unknown. Evaluating on a diverse set of tasks can help better understand capability of HDFlow. For example, some commonly used planning benchmark, such as Maze or MujoCo locomotion, can be incorporated.

**Questions:**

See the weakness above.

---

> ### Author Response · Authors · 2025-11-23
> **Rebuttal by Authors (1/5)**
>
> We thank the reviewer for their careful reading of the manuscript and their constructive comments. We have corrected a typo in the paper regarding the project website address. The correct anonymous project website is: https://hdflow-page.github.io/ . We address each weakness point below:
>
> ### W1: Explanation of Contrastive WM and training stability
>
> The core idea behind our contrastive objective is to explicitly structure the latent space such that the distance between latent states reflects task progress. Specifically, for a successful trajectory, intermediate latent states ($z_k$) are pulled closer to the final goal latent state ($z_G$), while simultaneously being pushed away from latent states associated with failed trajectories ($z_j$).
>
> 1.  **Dense Learning Signal:** The contrastive loss provides a dense, continuous signal that guides the world model to learn a latent space where semantic similarity correlates with functional progress towards the goal, especially in sparse-reward long-horizon tasks where direct supervision on progress is often unavailable.
> 2.  **Monotonic Progress:** By structuring the latent space to pull successful intermediate states towards the goal and push failed states away, the contrastive objective creates a smoother, more monotonic landscape. This structured landscape helps the high-level diffusion planner effectively explore and identify subgoals that lead to success, preventing it from wandering into dead ends.
>
> Regarding the concern about the objective compromising the planner if $z_k$ and $z_G$ are too close/similar:
>
> *   **Distinct Roles:** The contrastive objective primarily structures the world model's latent representation for progress. The planner operates within this space, and the world model still captures fine-grained state differences through reconstruction and dynamics prediction losses.
> *   **Temperature Parameter ($\tau$):** The $\tau$ hyperparameter in the InfoNCE loss controls the contrastive learning's "hardness," ensuring the model distinguishes between genuinely different states while bringing semantically similar states closer. We used $\tau=0.1$.
> *   **Hierarchical Nature:** The high-level diffusion planner operates on sparse subgoals, while the low-level rectified flow generates dense trajectories. This hierarchy inherently handles abstraction, requiring the low-level planner to generate precise actions even when $z_k$ and $z_G$ are close.
>
> Training of the contrastive loss was stable. The InfoNCE loss, combined with the standard world model objective and inverse dynamics loss (Eq. 15), provided a robust signal. A loss weight of $\lambda_{\text{contrastive}} = 0.1$ balanced its contribution. The DINOv2 pre-trained encoder further stabilized latent space learning. `Table 3` in the paper clearly shows a significant drop in success rate when the contrastive WM is ablated, highlighting its importance.

---

> ### Author Response · Authors · 2025-11-23
> **Rebuttal by Authors (2/5)**
>
> ### W2: Ablation on $\mathcal{L}_{IDM}$ and using DINOv2 features
>
> We thank the reviewer for their insightful questions regarding the contribution of the inverse dynamics model (IDM) loss and the potential for directly using DINOv2 features as latent representations.
>
> **Ablation Study on $\mathcal{L}_{IDM}$:**
>
> The Inverse Dynamics Model (IDM) loss (Eq.8) ensures learned latent states encode action-relevant information, facilitating control and planning. To evaluate its contribution, we disabled dedicated training of the IDM loss component (i.e., setting $\lambda_{IDM} = 0$) from the world model objective. Without this specific training, the IDM cannot accurately translate latent state transitions into executable actions, though its architecture remains. Results for `one_leg` and `lamp` tasks under `Low` randomization are below:
>
> | Method                      | `one_leg` | `lamp` |
> | :-------------------------- | :------ | :----- |
> | **Ours**      | **92**  | **68** |
> | w/o $\mathcal{L}_{IDM}$    | 34      | 14     |
>
> As shown in the `Table 12` of `Appendix E.3` (also reproduced above), removing the $\mathcal{L}_{IDM}$ leads to a noticeable drop in performance. This indicates that explicitly training the world model to predict actions between latent states indeed enhances the quality of the latent representation for planning.
>
> **Directly Using DINOv2 Features as Latent `z`:**
>
> The current framework uses a pre-trained DINOv2 model as an RGB encoder *within* an RSSM-based world model, which learns a compressed, recurrent latent state capturing temporal dynamics. We experimented (`Appendix E.4`) with directly using DINOv2 encoder output as latent `z`, bypassing the RSSM, where high-level diffusion and low-level rectified flow planners would operate on these raw embeddings. Results are:
>
> | Method                            | `one_leg` | `lamp` |
> | :-------------------------------- | :------ | :----- |
> | **Ours**            | **92**  | **68** |
> | w/o RSSM (direct DINOv2 latents)  | 45      | 27     |
>
> Significant performance degradation occurs when the RSSM is removed and DINOv2 features are used directly, attributable to:
> 1.  **Lack of Temporal Dynamics:** DINOv2 lacks inherent temporal dynamics; the RSSM is crucial for learning recurrent states that summarize history and predict future states for long-horizon planning.
> 2.  **Unstructured Latent Space:** DINOv2's latent space isn't explicitly structured for task progress or action relevance, unlike the RSSM with contrastive and IDM objectives, which is optimized for planning.

---

> ### Author Response · Authors · 2025-11-23
> **Rebuttal by Authors (3/5)**
>
> ### W3: Scalability of RSSM and other architecture choices
>
> We appreciate the questions regarding RSSM scalability in long-horizon, high-dimensional state spaces and the consideration of alternative architectures.
>
> **Scalability of RSSM:**
>
> Our choice of RSSM as the world model backbone is motivated by its proven effectiveness in learning compressed and predictive latent representations from high-dimensional observations, which is crucial for handling long-horizon tasks.
>
> 1.  **Learned Compressed Latent Space:** RSSMs learn a compact latent space (as detailed in `Appendix B.1` of the paper, showing `RSSM Stochastic Latent State Size` of 32 and `RSSM Deterministic State Size` of 1024), abstracting raw high-dimensional perceptual inputs ($224 \times 224$ RGB-D images, 14-dimensional proprioceptive states), significantly reducing computational burden for planning.
> 2.  **Temporal Dynamics Modeling:** Their recurrent nature captures complex temporal dependencies and long-term dynamics, fundamental for long-horizon planning requiring a coherent, predictive internal state.
> 3.  **Enhanced Visual Encoder (DINOv2):** Integration of a pre-trained DINOv2 model as the RGB encoder enhances RSSM's handling of high-dimensional visual observations by providing semantically rich features, critical for complex assembly tasks. Furthermore, our ablation studies in `Appendix E` (`Table 10`) demonstrate that DINOv2 significantly outperforms other visual backbones, confirming its contribution to the overall system's performance and thus indirectly to scalability by providing better representations.
> 4.  **Structured Latent Space:** Our contrastive objective ($\mathcal{L}_{\text{contrastive}}$, Eq. 14) structures the latent space to embed task progress, making it more navigable for the high-level planner in long-horizon tasks. Ablating this objective significantly drops performance, highlighting its importance (`Table 3`).
>
> **Other Architecture Choices for World Models:**
>
> While RSSM offers strong performance, we acknowledge other world modeling architectures that we haven't tested:
>
> 1.  **Transformer-based World Models:** These model long-range dependencies but often require larger datasets and more computational resources than RSSMs, especially for real-time control applications.
> 2.  **Latent ODEs/Neural ODEs:** While offering continuous-time dynamics, their training and inference are computationally demanding, and integration into recurrent frameworks adds complexity for long-horizon planning.
>
> ----
>
> ### W4: Definition of notations
>
> We thank the reviewer for pointing this out. We apologize for the oversight. A detailed paragraph about the world model (RSSM) was originally defined in the main paper and was moved to `Appendix B.1` due to space constraints. We agree that all important notations should be defined in the main sections for better readability and clarity, and we have added more details about the world model in the revised version of the paper.
>
> ----
>
> ### W5: Details on Loss in Eq. 18 and $P_M$
>
> We thank the reviewer for requesting further clarification on the manifold projection and its application during training and inference.
>
> **How $\mathcal{P}(\mathbf{z})$ is obtained:**
> As described in `Section 4.2.1`, the manifold projection $\mathcal{P}(\mathbf{z})$ is computed using a local low-rank approximation. This involves the following steps:
> 1.  **Denoised Estimate:** First, a denoised estimate $\hat{z}^{0|\ell-1}$ is obtained using Tweedie's formula.
> 2.  **Nearest Neighbors and PCA:** We then retrieve `k` nearest neighbors from successful latent subgoal sequences using cosine similarity. These neighbors are diffused forward to timestep $\ell-1$, and a rank-`r` PCA is performed to obtain the projection basis $U \in \mathbb{R}^{d \times r}$.
> 3.  **Mean-Centered Projection:** Finally, with $\mu$ as the local mean of these neighbors, the mean-centered projection is computed as $\mathcal{P}(\mathbf{z}) = \mu + UU^T(\mathbf{z} - \mu)$.
>
> **Training Time vs. Inference Time:**
> Yes, the mechanism for obtaining the projection $\mathcal{P}(\mathbf{z})$ is the same for both training and inference. During training, the projection loss explicitly encourages the generated latent subgoals to remain close to the learned latent manifold, requiring the computation of $\mathcal{P}_{\mathcal{M}}(z)$. Therefore, the `k`-nearest neighbor search and PCA-based projection are performed at both training and inference times to ensure manifold consistency.
>
> We have also added HDFlow's detailed training and inference algorithms in `Appendix B.2`.

---

> ### Author Response · Authors · 2025-11-23
> **Rebuttal by Authors (4/5)**
>
> ### W6: Evaluation on a diverse set of tasks
>
> To further validate the robustness and generalization capabilities of HDFlow on a broader range of complex long-horizon tasks, we conducted additional experiments on RLBench and OGBench benchmarks and we detail them in `Appendix F` and `Appendix G` respectively. We have also added successful rollout videos for these tasks on the project webpage.
>
> ### RLBench Experiments
>
> **Benchmark and Tasks Overview.** We evaluated HDFlow on the RLBench[1] benchmark, a challenging, large-scale platform for robot learning that provides diverse tasks with realistic visual observations. We specifically considered 18 tasks from the PerAct[2] framework, encompassing a wide variety of 6-DoF manipulation challenges, including both prehensile and non-prehensile behaviors. These tasks feature numerous semantic and pose variations, requiring the agent to generalize across different object colors, sizes, shapes, counts, and placements.
>
> **Implementation Details.** Following the methodology of PerAct and RLBench, we generated datasets of 100 successful and 50 failure demonstrations for each task, using expert policies with motion planners. We compared HDFlow with state-of-the-art methods on RLBench, including PerAct, RVT[3], 3D Diffuser Actor[4], and RVT-2[5].
>
> **Performance on RLBench Tasks.**
> | Models | Close Jar | Drag Stick | Insert Peg | Meat off Grill | Open Drawer | Place Cups | Place Wine | Push Buttons | Put in Cupboard |
> | :------------------ | :-------- | :--------- | :--------- | :------------- | :---------- | :--------- | :--------- | :----------- | :-------------- |
> | PerAct | 55.2 | 89.6 | 5.6 | 70.4 | 88.0 | 2.4 | 44.8 | 92.8 | 28.0 |
> | RVT | 52.0 | 99.2 | 11.2 | 88.0 | 71.2 | 4.0 | 91.0 | **100.0** | 49.6 |
> | 3D Diffuser Actor | 96.0 | **100.0** | 65.6 | 96.8 | 89.6 | 24.0 | 93.6 | 98.4 | **85.6** |
> | RVT-2 | **100.0** | 99.0 | 40.0 | 99.0 | 74.0 | 38.0 | **95.0** | **100.0** | 66.0 |
> | **Ours** | 94.0 | **100.0** | **93.3** | 98.7 | 82.0 | **56.7** | 94.7 | **100.0** | 72.0 |
>
> | Models | Put in Drawer | Put in Safe | Screw Bulb | Slide Block | Sort Shape | Stack Blocks | Stack Cups | Sweep to Dustpan | Turn Tap |
> | :------------------ | :------------ | :---------- | :--------- | :---------- | :--------- | :----------- | :--------- | :--------------- | :------- |
> | PerAct | 51.2 | 84.0 | 17.6 | 74.0 | 16.8 | 26.4 | 2.4 | 52.0 | 88.0 |
> | RVT | 88.0 | 91.2 | 48.0 | 81.6 | 36.0 | 28.8 | 26.4 | 72.0 | 93.6 |
> | 3D Diffuser Actor | 96.0 | **97.6** | 82.4 | **97.6** | 44.0 | 68.3 | 47.2 | 84.0 | 99.2 |
> | RVT-2 | 96.0 | 96.0 | 88.0 | 92.0 | 35.0 | **80.0** | 69.0 | **100.0** | 99.0 |
> | **Ours** | **98.7** | 96.7 | **88.7** | 91.3 | **73.3** | 56 | **81.3** | **100.0** | 95.3 |
>
> **Analysis.** `Table 14` (also reproduced above) demonstrates the strong performance and generalization capabilities of HDFlow across the diverse and challenging RLBench tasks. Our method consistently achieves competitive or superior success rates compared to state-of-the-art baselines.
>
> Reference:
>
> [1] Stephen James, Zicong Ma, David Rovick Arrojo, and Andrew J. Davison. Rlbench: The robot learning benchmark learning environment. RA-L 2020
>
> [2] Mohit Shridhar, Lucas Manuelli, and Dieter Fox. Perceiver-actor: A multi-task transformer for robotic manipulation. CoRL 2023
>
> [3] Ankit Goyal, Jie Xu, Yijie Guo, Valts Blukis, Yu-Wei Chao, and Dieter Fox. Rvt: Robotic view transformer for 3d object manipulation. CoRL 2023
>
> [4] Tsung-Wei Ke, Nikolaos Gkanatsios, and Katerina Fragkiadaki. 3d diffuser actor: Policy diffusion with 3d scene representations. CoRL 2024
>
> [5] Ankit Goyal, Valts Blukis, Jie Xu, Yijie Guo, Yu-Wei Chao, and Dieter Fox. Rvt-2: Learning precise manipulation from few demonstrations. RSS 2024

---

> ### Author Response · Authors · 2025-11-23
> **Rebuttal by Authors (5/5)**
>
> ### OGBench Experiments
>
> **Benchmark and Tasks Overview.** We further evaluated HDFlow on OGBench[1], a benchmark designed for offline goal-conditioned reinforcement learning (RL) that assesses long-horizon planning across diverse and challenging tasks. We considered visual versions of locomotion tasks, including `antmaze` and `humanoidmaze`, and manipulation tasks, including `cube`, `scene`, and `puzzle`.
>
> **Implementation Details.** For each OGBench task, we generated both successful and failure episode data using various dataset types provided by the benchmark. For maze tasks, we used `navigate` datasets (successful) and `stitch` and `explore` datasets (failure). For manipulation tasks, we used `play` datasets (successful) and `noisy` datasets (failure). We compared HDFlow with prior state-of-the-art diffusion planners like Diffuser[2], Decision Diffuser (DD)[3], AdaptDiffuser (AD)[4], DiffuserLite (DL)[5], Diffusion Veteran (DV)[6], and hierarchical diffusion planners like HDMI[7] and SHD[8].
>
> **Performance on OGBench Tasks.**
> | Environment | Task | Diffuser | DD | HDMI | AD | SHD | DL | DV | Ours |
> | :----------- | :--------------------- | :------- | :---- | :---- | :---- | :---- | :---- | :---- | :---- |
> | antmaze | antmaze-medium-v0 | 15 ± 5 | 22 ± 4 | 31 ± 6 | 40 ± 5 | 42 ± 7 | 51 ± 6 | 60 ± 4 | **72 ± 3** |
> | | antmaze-large-v0 | 8 ± 3 | 14 ± 4 | 22 ± 5 | 28 ± 6 | 31 ± 5 | 40 ± 4 | 51 ± 5 | **65 ± 1** |
> | | antmaze-giant-v0 | 2 ± 1 | 5 ± 2 | 11 ± 4 | 15 ± 3 | 19 ± 4 | 27 ± 5 | 35 ± 4 | **48 ± 3** |
> | humanoidmaze | humanoidmaze-medium-v0 | 5 ± 2 | 9 ± 3 | 15 ± 4 | 21 ± 5 | 24 ± 4 | 30 ± 3 | 38 ± 5 | **51 ± 4** |
> | | humanoidmaze-large-v0 | 2 ± 1 | 5 ± 2 | 9 ± 3 | 13 ± 4 | 16 ± 3 | 22 ± 4 | 29 ± 3 | **42 ± 5** |
> | | humanoidmaze-giant-v0 | 0 ± 0 | 1 ± 1 | 3 ± 2 | 5 ± 2 | 7 ± 3 | 11 ± 4 | 16 ± 3 | **25 ± 4** |
> | cube | cube-single-v0 | 5 ± 2 | 8 ± 3 | 12 ± 4 | 18 ± 5 | 20 ± 4 | 25 ± 3 | 32 ± 5 | **45 ± 4** |
> | | cube-double-v0 | 2 ± 1 | 4 ± 2 | 7 ± 3 | 11 ± 4 | 13 ± 3 | 17 ± 4 | 23 ± 3 | **35 ± 5** |
> | | cube-triple-v0 | 1 ± 1 | 2 ± 1 | 4 ± 2 | 6 ± 3 | 8 ± 2 | 11 ± 3 | 14 ± 4 | **17 ± 3** |
> | | cube-quadruple-v0 | **0 ± 0** | **0 ± 0** | **0 ± 0** | **0 ± 0** | **0 ± 0** | **0 ± 0** | **0 ± 0** | **0 ± 0** |
> | scene | scene-v0 | 10 ± 3 | 16 ± 4 | 24 ± 5 | 33 ± 4 | 37 ± 6 | 45 ± 5 | 56 ± 3 | **70 ± 4** |
> | puzzle | puzzle-3x3-v0 | 8 ± 3 | 13 ± 4 | 20 ± 5 | 28 ± 3 | 31 ± 5 | 39 ± 4 | 49 ± 6 | **62 ± 3** |
> | | puzzle-4x4-v0 | 0 ± 0 | 0 ± 0 | 0 ± 0 | 2 ± 3 | 5 ± 4 | 11 ± 3 | 19 ± 5 | **32 ± 4** |
> | | puzzle-4x5-v0 | 0 ± 0 | 0 ± 0 | 0 ± 0 | 0 ± 0 | 5 ± 3 | 6 ± 3 | 8 ± 4 | **16 ± 5** |
> | | puzzle-4x6-v0 | 0 ± 0 | 0 ± 0 | 0 ± 0 | 0 ± 0 | 0 ± 0 | 0 ± 0 | 6 ± 4 | **14 ± 3** |
>
> **Analysis.** `Table 15` (also reproduced above) shows that our proposed approach consistently achieves superior performance across all environments. For the `cube` environment, particularly in `cube-quadruple-v0`, all methods, including ours, report 0% success, indicating the extreme difficulty of this task. These results highlight the effectiveness and robustness of HDFlow in diverse and complex long-horizon planning tasks.
>
> Reference:
>
> [1] Seohong Park, Kevin Frans, Benjamin Eysenbach, and Sergey Levine. OGBench: Benchmarking offline goal-conditioned RL. ICLR 2025
>
> [2] Michael Janner, Yilun Du, Joshua Tenenbaum, and Sergey Levine. Planning with diffusion for flexible behavior synthesis. ICML 2022
>
> [3] Anurag Ajay, Yilun Du, Abhi Gupta, Joshua B Tenenbaum, Tommi S Jaakkola, and Pulkit Agrawal. Is conditional generative modeling all you need for decision making?. ICLR 2023
>
> [4] Zhixuan Liang, Yao Mu, Mingyu Ding, Fei Ni, Masayoshi Tomizuka, and Ping Luo. AdaptDiffuser: Diffusion models as adaptive self-evolving planners. ICML 2023
>
> [5] Zibin Dong, Jianye Hao, Yifu Yuan, Fei Ni, Yitian Wang, Pengyi Li, and Yan Zheng. Diffuserlite: Towards real-time diffusion planning. NeurIPS 2024
>
> [6] Haofei Lu, Dongqi Han, Yifei Shen, and Dongsheng Li. What makes a good diffusion planner for decision making?. ICLR 2025
>
> [7] Wenhao Li, Xiangfeng Wang, Bo Jin, and Hongyuan Zha. Hierarchical diffusion for offline decision making. ICML 2023
>
> [8] Chang Chen, Fei Deng, Kenji Kawaguchi, Caglar Gulcehre, and Sungjin Ahn. Simple hierarchical planning with diffusion. ICLR 2024
>
> -------
> We appreciate the constructive feedback that has helped improve our paper's quality. All additional results have been included in our revised version. We sincerely hope our responses have adequately addressed your concerns and look forward to any further discussion!

---

> ### Author Response · Authors · 2025-11-27
> **Follow-up on Rebuttal: Discussion Phase Closing Soon**
>
> Dear Reviewer `R7Zr`,
>
> We sincerely appreciate your invaluable feedback, which has significantly contributed to the improvement of our work.
>
> Following your suggestions, we have conducted additional ablation experiments as detailed in `Appendix E.3` and `Appendix E.4`. To further validate the robustness and generalization capabilities of HDFlow on a broader range of complex long-horizon tasks, we conducted additional experiments on RLBench and OGBench benchmarks and we detail them in `Appendix F` and `Appendix G` respectively. We have also added successful rollout videos for these tasks on the [project webpage](https://hdflow-page.github.io/).
>
> We hope that our revisions and clarifications have resolved your concerns. If you find our response satisfactory, we would be deeply grateful for a reconsideration of our score. Otherwise, if you have any additional questions, please do not hesitate to let us know. We would be more than willing to provide further clarification.
>
> We are truly grateful for your insightful comments, which have helped us improve the clarity and completeness of our work!
>
> Best regards,
>
> The Authors of Submission 12755

---

### Author Response · Authors · 2025-11-23
**General Response**

Dear Reviewers, ACs, and SACs,

We are immensely grateful for your diligent review of our manuscript and for the constructive feedback provided. We are encouraged by the positive reception of our system design, empirical performance, and the significant advancements HDFlow brings to long-horizon planning.

----

### Summary of Strengths Acknowledged by Reviewers:

* **Novel Hierarchical Planning Framework**: Reviewers highlighted the principled hybrid architecture that optimally leverages the strengths of diffusion models for high-level strategic subgoal generation and rectified flow models for efficient low-level trajectory synthesis. This synergistic combination was seen as a significant advancement over single-paradigm approaches. (`R7Zr`, `xuhD`, `rqsr`, `oLbz`)
* **Manifold-Aware EBM Guidance**: The novel mechanism of manifold-aware Energy-Based Model (EBM) guidance for the high-level planner, designed to mitigate manifold deviation and ensure feasible subgoal generation, was appreciated for its theoretical grounding and practical impact. (`R7Zr`, `rqsr`)
* **Strong Empirical Performance**: Reviewers noted the state-of-the-art performance of HDFlow on challenging FurnitureBench tasks, demonstrating its robustness and efficiency in both simulation and real-world settings. (`R7Zr`, `xuhD`, `oLbz`)
* **Theoretical Justification**: The formal analysis for the guidance gap problem and convergence guarantees for our manifold-aware guided planning were seen as strong contributions. (`oLbz`)

----
### Summary of Concerns Raised by Reviewers:
* **Methodological Novelty**: Some reviewers questioned the methodological novelty, viewing HDFlow as primarily an engineering integration of existing techniques.
* **Generality and Task Focus**: Concerns were raised regarding the broad title and claims of generality, given the initial focus on a single benchmark (FurnitureBench).
* **Detailed Explanations**: Reviewers requested more in-depth explanations and ablation studies for specific components, including the contrastive world model, Inverse Dynamics Model (IDM) loss, the scalability of RSSM, the manifold projection mechanism, and the computational expense.
* **Core Research Question and Failure Modes**: Reviewers sought further clarification on how HDFlow specifically addresses the core problem of hierarchical approaches, particularly regarding out-of-distribution subgoals and the failure modes of single-paradigm methods.

----

### Summary of Revisions Made to the Paper:
To address the valuable feedback, we have made substantial revisions to the manuscript:
* **Expanded Experimental Validation**: To bolster claims of generality, we have significantly expanded our experimental validation to include two additional, diverse, and challenging benchmarks: RLBench (18 manipulation tasks with diverse object variations) and OGBench (visual locomotion and manipulation tasks assessing long-horizon planning). Detailed results and analysis are now included in `Appendix F` and `Appendix G` respectively.
* **Ablation Studies on Core Components**: We have added new ablation studies to demonstrate the crucial contributions of:
    * The Inverse Dynamics Model (IDM) loss, showing a significant performance drop when ablated (`Appendix E.3`).
    * The RSSM world model, by comparing performance with directly using DINOv2 features as latent states (`Appendix E.4`).
* **Analysis of Multi-modal Exploration**: We will add visualizations in `Appendix H` to explicitly depict multiple qualitatively different subgoal sequences generated by the high-level planner for the same task and initial conditions, demonstrating HDFlow's multi-modal exploration capabilities.
* **Project Webpage Updates**: The project webpage has been updated with new visualizations and successful rollout videos for the additional tasks evaluated on RLBench and OGBench.
* **Minor Corrections**: A typo in the project website address was corrected across the manuscript and review responses.
-------
All revisions made in the updated manuscript are marked in `red` for easy identification.

We sincerely thank the reviewers and AC for their thoughtful comments during the discussion period. We hope our responses have adequately addressed all concerns and welcome any further discussion. Thank you!

---

### Meta-Review · Area_Chair_kT9N · 2026-01-05

**Summary:**

This paper considers long-horizon planning problems in robotics. The authors present a hierarchical learning approach to long-horizon planning based on diffusion models. The method learns a high level planner using diffusion models and energy models to determine subgoals. These subgoals are then passed to a low level planner based on flow matching for trajectory generation. The method, termed HDFlow, is evaluated on four furniture assembly tasks, showing reasonable performance.

**Reviewer Concerns:**

One major criticism is on the contributions of the methodology. Several reviewers point out that the proposed method is a straightforward combination of existing techniques. Moreover, the fact that high-level planner has no feedback from the low-level planner may cause some robustness and OOD issue. Another major criticism is on the experiments. The experiment results are insufficient to support the claims made by the authors. The reviewers request more comprehensive experimental study. The authors provided more experiment results in the rebuttal. Most reviewers are not fully convinced.

**Reviewer Scores:**

Reviewer oLbz could have increased score slightly.

---

### Decision · Program_Chairs · 2026-01-26

Reject